# Muscle function and homeostasis require cytokine inhibition of AKT activity in *Drosophila*

Katrin Kierdorf[1,2†‡§]*, Fabian Hersperger[3,4], Jessica Sharrock[1,2#], Crystal M Vincent[1,2], Pinar Ustaoglu[1,2], Jiawen Dou[1], Attila Gyoergy[5], Olaf Groß[3,6,7], Daria E Siekhaus[5], Marc S Dionne[1]*

[1]MRC Centre for Molecular Bacteriology and Infection, Imperial College London, London, United Kingdom; [2]Department of Life Sciences, Imperial College London, London, United Kingdom; [3]Institute of Neuropathology, Faculty of Medicine, University of Freiburg, Freiburg, Germany; [4]Faculty of Biology, University of Freiburg, Freiburg, Germany; [5]Institute of Science and Technology, Klosterneuburg, Austria; [6]Centre for Integrative Biological Signalling Studies (CIBSS), University of Freiburg, Freiburg, Germany; [7]Center for Basics in NeuroModulation (NeuroModulBasics), Faculty of Medicine, University of Freiburg, Freiburg, Germany

*For correspondence:
katrin.kierdorf@uniklinik-freiburg.de (KK);
m.dionne@imperial.ac.uk (MSD)

Present address: [†]Institute of Neuropathology, Faculty of Medicine, University of Freiburg, Breisacherstraße, Germany; [‡]Centre for Integrative Biological Signalling Studies(CIBSS), University of Freiburg, Freiburg, Germany; [§]Center for Basics in NeuroModulation (NeuroModulBasics), Faculty of Medicine, University of Freiburg, Freiburg, Germany; [#]Immunology Program, Memorial Sloan-Kettering Cancer Center, New York, United States

Competing interests: The authors declare that no competing interests exist.

**Abstract** *Unpaired* ligands are secreted signals that act via a GP130-like receptor, *domeless*, to activate JAK/STAT signalling in *Drosophila*. Like many mammalian cytokines, *unpaireds* can be activated by infection and other stresses and can promote insulin resistance in target tissues. However, the importance of this effect in non-inflammatory physiology is unknown. Here, we identify a requirement for *unpaired*-JAK signalling as a metabolic regulator in healthy adult *Drosophila* muscle. Adult muscles show basal JAK-STAT signalling activity in the absence of any immune challenge. Plasmatocytes (*Drosophila* macrophages) are an important source of this tonic signal. Loss of the *dome* receptor on adult muscles significantly reduces lifespan and causes local and systemic metabolic pathology. These pathologies result from hyperactivation of AKT and consequent deregulation of metabolism. Thus, we identify a cytokine signal that must be received in muscle to control AKT activity and metabolic homeostasis.

## Introduction

JAK/STAT activating signals are critical regulators of many biological processes in animals. Originally described mainly in immune contexts, it has increasingly become clear that JAK/STAT signalling is also central to metabolic regulation in many tissues (*Dodington et al., 2018*; *Villarino et al., 2017*). One common consequence of activation of JAK/STAT pathways in inflammatory contexts is insulin resistance in target tissues, including muscle (*Kim et al., 2013*; *Mashili et al., 2013*). However, it is difficult to describe a general metabolic interaction between JAK/STAT and insulin signalling in mammals, due to different effects at different developmental stages, differences between acute and chronic actions, and the large number of JAKs and STATs present in mammalian genomes (*Dodington et al., 2018*; *Mavalli et al., 2010*; *Nieto-Vazquez et al., 2008*; *Vijayakumar et al., 2013*).

The fruit fly *Drosophila melanogaster* has a single, well-conserved JAK/STAT signalling pathway. The *unpaired (upd)* genes *upd1-3* encode the three known ligands for this pathway; they signal by binding to a single common GP130-like receptor, encoded by *domeless* (*dome*) (*Agaisse et al., 2003*; *Brown et al., 2001*; *Chen et al., 2002*). Upon ligand binding, the single JAK tyrosine kinase in *Drosophila*, encoded by *hopscotch* (*hop*), is activated; Hop then activates the single known STAT,

**eLife digest** The immune system helps animals fend off infections, but it also has a role in controlling the body's metabolism – that is, the chemical reactions that sustain life. For instance, in fruit flies, high-fat diets can trigger the immune system, which results in cells becoming resistant to the hormone insulin and not being able to process sugar properly; this in turn leads to problems in sugar levels and shorter lifespans. This mechanism involves the release of an immune signal called *unpaired-3 (upd3)*, which then binds to a receptor known as *dome*. Yet, it was unclear how exactly the immune system and metabolism work together, and whether their interactions are also important in flies on a normal diet.

To investigate, Kierdorf et al. stopped the activity of the *dome* receptor in the muscles of healthy flies. This led to an increase in the activity of the enzyme AKT, a protein critical to relay insulin-type signals inside the cell. As a result, insulin signaling was hyperactivated in the tissue, leading to decreased muscle function, unhealthy changes in how energy was stored and spent, and ultimately, a shorter life for the insects. Further experiments also identified blood cells called plasmatocytes (the flies' equivalent of certain human immune cells) as a key source of the *upd* signal.

The findings by Kierdorf et al. shed a light on the fact that, even in healthy animals, complex interactions are required between the immune system and the metabolism. Further investigations will reveal if other body parts besides muscles rely on similar connections.

STAT92E, which functions as a homodimer (*Binari and Perrimon, 1994*; *Chen et al., 2002*; *Hou et al., 1996*; *Yan et al., 1996*). This signalling pathway plays a wide variety of functions, including segmentation of the early embryo, regulation of hematopoiesis, maintenance and differentiation of stem cells in the gut, and immune modulation (*Amoyel and Bach, 2012*; *Myllymäki and Rämet, 2014*). Importantly, several recent studies indicate roles for *upd* cytokines in metabolic regulation; for example, *upd*s are important nutrient-responsive signals in the adult fly (*Beshel et al., 2017*; *Rajan and Perrimon, 2012*; *Woodcock et al., 2015*; *Zhao and Karpac, 2017*).

Here, we identify a physiological requirement for Dome signalling in adult muscle. We observe that adult muscles show significant JAK/STAT signalling activity in the absence of obvious immune challenge. Plasmatocytes are a source of this signal. Inactivation of *dome* on adult muscles significantly reduces lifespan and causes muscular pathology and physiological dysfunction; these result from remarkably strong AKT hyperactivation and consequent dysregulation of metabolism. We thus describe a new role for JAK/STAT signalling in adult *Drosophila* muscle with critical importance in healthy metabolic regulation.

## Results

### *Dome* is required in adult muscle

To find physiological functions of JAK/STAT signalling in the adult fly, we identified tissues with basal JAK/STAT pathway activity using a STAT-responsive GFP reporter (*10xSTAT92E-GFP*) (*Bach et al., 2007*). The strongest reporter activity we observed was in legs and thorax. We examined flies also carrying a muscle myosin heavy chain RFP reporter (*MHC-RFP*) and observed co-localization of GFP and RFP expression in the muscles of the legs, thorax and body wall (*Figure 1—figure supplement 1*). We observed strong, somewhat heterogeneous reporter expression in all the muscles of the thorax and the legs, with strong expression in various leg and jump muscles and apparently weaker expression throughout the body wall muscles and indirect flight muscles (*Figure 1A*). *dome* encodes the only known *Drosophila* STAT-activating receptor. To investigate the physiological role of this signal, we expressed *dome*$^\Delta$, a dominant-negative version of Dome lacking the intracellular signalling domain, with a temperature-inducible muscle specific driver line (*w;tubulin-Gal80$^{ts}$;24B-Gal4*) (*Figure 1—figure supplement 1*) (*Brown et al., 2001*). Controls (*24B-Gal80$^{ts}$/+*) and experimental flies (*24B-Gal80$^{ts}$ > dome$^\Delta$*) were raised at 18˚C until eclosion to permit Dome activity during development. Flies were then shifted to 29˚C to inhibit Dome activity and their lifespan was monitored. Flies with Dome signalling inhibited in adult muscles were short-lived (*Figure 1B*, *Figure 1—figure supplement 1*). This effect was also observed, more weakly, in flies kept at 25˚C (*Figure 1—figure*

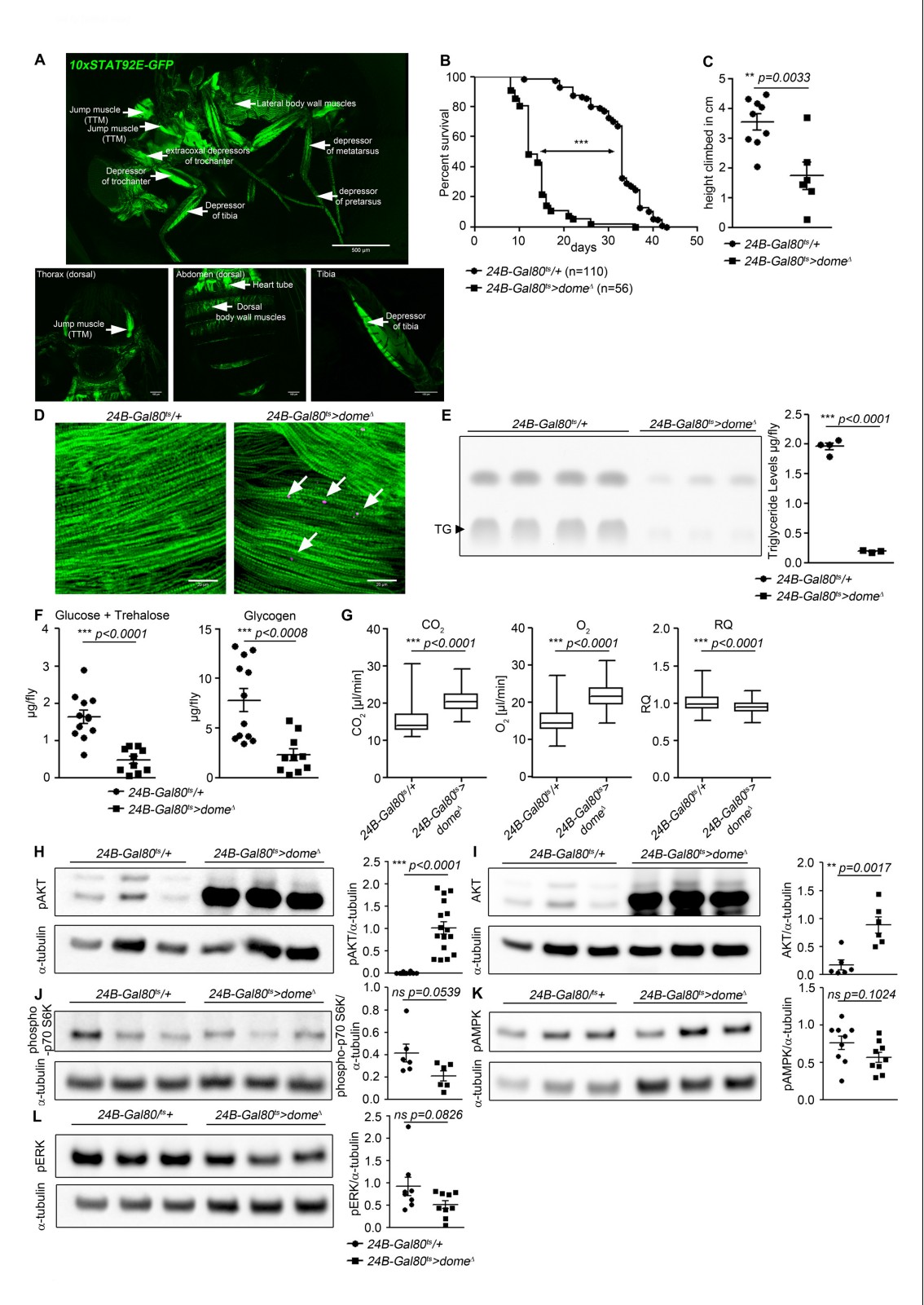

**Figure 1.** Dome inhibition in adult muscle reduces lifespan, disrupts homeostasis, and causes AKT hyperactivation. (**A**) STAT activity in different muscles in 10xSTAT92E-GFP reporter fly. One fly out of 5 shown. Upper panel: lateral view, Scale bar = 500 µm. Lower panels: dorsal thorax (left); dorsal abdomen (middle); tibia (right), Scale bar = 100 µm. (**B**) Lifespan of *24B-Gal80$^{ts}$/+* and *24B-Gal80$^{ts}$ > dome$^\Delta$* at 29°C. Log-Rank test: $\chi^2 = 166$, ***p<0.0001; Wilcoxon test: $\chi^2 = 157.7$, ***p<0.0001. (**C**) Negative geotaxis assay of 14-day-old *24B-Gal80$^{ts}$/+* and *24B-Gal80$^{ts}$ > dome$^\Delta$* flies. Points

*Figure 1 continued on next page*

*Figure 1 continued*

represent mean height climbed in individual vials (~20 flies/vial), pooled from three independent experiments. Unpaired T-test: **p=0.0033. (D) Muscle (Phalloidin) and neutral lipid (LipidTox) of thorax samples from 14-day-old *24B-Gal80^{ts}/+* and *24B-Gal80^{ts} > dome^{Δ}* flies. One representative fly per genotype is shown of six analysed. Scale bar = 50 μm. (E) Thin layer chromatography (TLC) of triglycerides in 7-day-old *24B-Gal80^{ts}/+* and *24B-Gal80^{ts} > dome^{Δ}* flies, n = 3–4 per genotype. One experiment of two is shown. Unpaired T-Test: ***p<0.0001. (F) Glucose and trehalose (left) and glycogen (right) in 7-day-old *24B-Gal80^{ts}/+* and *24B-Gal80^{ts} > dome^{Δ}* flies, pooled from two independent experiments. Unpaired T-Test (Glucose +Trehalose): ***p<0.0001* and unpaired T-Test (Glycogen): ***p<0.0001. (G) $CO_2$ produced, $O_2$ consumed, and RQ of 7-day-old *24B-Gal80^{ts}/+* and *24B-Gal80^{ts} > dome^{Δ}* flies. Box plots show data from one representative experiment of three, with data collected from a 24 hr measurement pooled from 3 to 4 tubes per genotype with 10 flies/tube. P values from Mann-Whitney test. (H–L) Western blots of leg protein from 14-day-old *24B-Gal80^{ts}/+* and *24B-Gal80^{ts} > dome^{Δ}* flies. (H) Phospho-AKT (S505). One experiment of four is shown. Unpaired T-Test: ***p<0.0001. (I) Total AKT. One experiment of two is shown. Unpaired T-Test: **p=0.0017. (J) Phospho-p70 S6K (T398). One experiment of two is shown. Unpaired T-Test: ns p=0.0539. (K) Phospho-AMPKα (T173). One experiment of three is shown. Unpaired T-Test: ns p=0.1024. (L) Phospho-ERK (T202/Y204). One experiment of three is shown. Unpaired T-Test: ns p=0.0826.

The online version of this article includes the following figure supplement(s) for figure 1:

**Figure supplement 1.** Further characterisation of the requirement for *dome* in adult muscle.

*supplement 1*). Upd-JAK-STAT signalling is important to maintain gut integrity, and defects in gut integrity often precede death in *Drosophila*; however, our flies did not exhibit loss of gut integrity (*Figure 1—figure supplement 1*) (*Jiang et al., 2009*; *Rera et al., 2012*). To determine whether Dome inhibition caused meaningful physiological dysfunction, we assayed climbing activity in *24B-Gal80^{ts}/+* control flies and *24B-Gal80^{ts} > dome^{Δ}* flies. *24B-Gal80^{ts} > dome^{Δ}* flies showed significantly impaired climbing compared to controls (*Figure 1C*). Adult muscle-specific expression of *dome^{Δ}* with a second Gal4 line (*w;tub-Gal80^{ts};Mef2-Gal4*) gave a similar reduction in lifespan and decline in climbing activity, confirming that the defect resulted from a requirement for Dome activity in muscle (*Figure 1—figure supplement 1*).

Impaired muscle function is sometimes accompanied by lipid accumulation (*Baik et al., 2017*). Therefore, we stained thorax muscles with the neutral lipid dye LipidTox. In 14 day old flies, we detected numerous small neutral lipid inclusions in several muscles, including the large jump muscle (TTM), of *24B-Gal80^{ts} > dome^{Δ}* flies (*Figure 1D*).

## Muscle *dome* activity is required for normal systemic homeostasis

Having observed lipid inclusions in adult muscles, we analysed the systemic metabolic state of *24B-Gal80^{ts} > dome^{Δ}* flies. We observed significant reductions in total triglyceride, glycogen and free sugar (glucose + trehalose) in these animals (*Figure 1E,F*).

Reduced free sugar could result from increased cellular sugar uptake. Increased uptake of sugars by peripheral tissues should be reflected in increased metabolic stores or metabolic rate. Since metabolic stores were decreased in our flies, we tested metabolic rate by measuring respiration. $CO_2$ production and $O_2$ consumption were both significantly increased in *24B-Gal80^{ts} > dome^{Δ}* flies, indicating an overall increase in metabolic rate (*Figure 1G*). *dome* acts via *hop* to regulate AKT activity with little effect on other nutrient signalling pathways.

The observed metabolic changes imply differences in activity of nutrient-regulated signalling pathways in *24B-Gal80^{ts} > dome^{Δ}* flies. Several signalling pathways respond to nutrients, or their absence, to coordinate energy consumption and storage (*Britton et al., 2002*; *Lizcano et al., 2003*; *Ulgherait et al., 2014*). Of these, insulin signalling via AKT is the primary driver of sugar uptake by peripheral tissues.

We examined the activity of these signalling mechanisms in legs (a tissue source strongly enriched in muscle) from *24B-Gal80^{ts} > dome^{Δ}* flies. We found an extremely strong increase in abundance of the 60 kDa form of total and activated (S505-phosphorylated) AKT (*Figure 1H,I*). This change was also seen in legs from *Mef2-Gal80^{ts} > dome^{Δ}* flies, confirming that *dome* functions in muscles (*Figure 1—figure supplement 1*). We also saw this effect in flies carrying a different insertion of the *dome^{Δ}* transgene, under the control of a third muscle-specific driver, *MHC-Gal4*, though the effect was weaker; the weakness of this effect may be a consequence of the fact that the MHC-Gal4 driver is not expressed in visceral muscle (*Bland et al., 2010*) (*Figure 1—figure supplement 1*). These *MHC-Gal4 >dome^{Δ} (II)* animals were also short-lived relative to controls (*Figure 1—figure supplement 1*).

Elevated total AKT could result from increased transcript abundance or changes in protein production or stability. We distinguished between these possibilities by assaying *Akt1* mRNA; *Akt1* transcript levels were elevated in *24B-Gal80$^{ts}$ > dome$^\Delta$* muscle, but only by about 75%, suggesting that the large effect on AKT protein abundance must be, at least in part, post-transcriptional (*Figure 1—figure supplement 1*). Similarly, AKT hyperactivation could be driven by insulin-like peptide overexpression; however, we assayed the expression of *Ilp2-7* in whole flies and observed that none of these peptides were significantly overexpressed (*Figure 1—figure supplement 1*). Next, we analysed if the feeding behaviour is affected by the muscle-specific *dome* loss, but we could not find a difference in food uptake in *24B-Gal80$^{ts}$ > dome$^\Delta$* flies compared to controls (*Figure 1—figure supplement 1*).

Unlike AKT, the amino-acid-responsive TORC1/S6K and the starvation-responsive AMPK pathway showed no significant difference in activity in *24B-Gal80$^{ts}$ > dome$^\Delta$* flies (*Figure 1K,L*). However, flies with AMPK knocked down in muscle did exhibit mild AKT hyperactivation (*Figure 2—figure supplement 1*).

To identify signalling mediators acting between Dome and AKT, we first tested activity of the MAPK-ERK pathway, which can act downstream of the JAK kinase Hop (*Luo et al., 2002*). We found an insignificant reduction in ERK activity in *24B-Gal80$^{ts}$ > dome$^\Delta$* flies (*Figure 1M*). We then assayed survival and AKT activity in flies with *hop* (JAK), *Dsor1* (MEK) and *rl* (ERK) knocked down in adult muscle. *rl* and *Dsor1* knockdown gave mild or no effect on survival and pAKT (*Figure 2—figure supplement 1*). In contrast, *hop* knockdown gave a mild phenocopy of *dome$^\Delta$* with regards to survival and pAKT (*Figure 2—figure supplement 1*).

We further analysed the requirement for *hop* in muscle *dome* signalling by placing *24B-Gal80$^{ts}$ > dome$^\Delta$* on a genetic background carrying the viable gain-of-function allele *hop$^{Tum-l}$*. Flies carrying *hop$^{Tum-l}$* alone exhibited no change in lifespan, AKT phosphorylation, or muscle lipid deposition (*Figure 2A–C*). However, *hop$^{Tum-l}$* completely rescued lifespan and pAKT levels in *24B-Gal80$^{ts}$ > dome$^\Delta$* flies (*Figure 2D,E*), indicating that the physiological activity of muscle Dome is mediated via Hop and that signal is required, but not sufficient, to control muscle AKT activity.

## Increased AKT activity causes the effects of *dome* inhibition

The phenotype of *24B-Gal80$^{ts}$ > dome$^\Delta$* flies is similar to that previously described in flies with loss of function in *Pten* or *foxo* (*Demontis and Perrimon, 2010*; *Mensah et al., 2015*), suggesting that AKT hyperactivation might cause the *dome* loss of function phenotype; however, to our knowledge, direct activation of muscle AKT had not previously been analysed. We generated flies with inducible expression of activated AKT (*myr-AKT*) in adult muscles (*w;tubulin-Gal80$^{ts}$/+;24B-Gal4/UAS-myr-AKT* [*24B-Gal80$^{ts}$ > myr-AKT*]) (*Stocker et al., 2002*). These animals phenocopied *24B-Gal80$^{ts}$ > dome$^\Delta$* flies with regards to lifespan, climbing activity, metabolite levels, metabolic rate, and muscle lipid deposition (*Figure 3A–F*).

We concluded that AKT hyperactivation could cause the pathologies seen in *24B-Gal80$^{ts}$ > dome$^\Delta$* flies. Therefore, we tested whether reducing AKT activity could rescue *24B-Gal80$^{ts}$ > dome$^\Delta$* flies. We generated flies carrying muscle-specific inducible dominant negative dome (*UAS-dome$^\Delta$*) with dsRNA against *Akt1* (*UAS-AKT-IR*). These flies showed significantly longer lifespan than *24B-Gal80$^{ts}$ > dome$^\Delta$* and *24B-Gal80$^{ts}$ > AKT* IR flies, similar to all control genotypes analyzed (*Figure 3G*). Dome and AKT antagonism synergised to control the mRNA level of *dome* itself, further suggesting strong mutual antagonism between these pathways (*Figure 3—figure supplement 1*).

AKT hyperactivation should reduce FOXO transcriptional activity. To test whether this loss of FOXO activity caused some of the pathologies observed in *24B-Gal80$^{ts}$ > dome$^\Delta$* flies, we increased *foxo* gene dosage by combining *24B-Gal80$^{ts}$ > dome$^\Delta$* with a transgene carrying a FOXO-GFP fusion protein under the control of the endogenous *foxo* regulatory regions. These animals exhibited rescue of physiological defects and lifespan compared to *24B-Gal80$^{ts}$ > dome$^\Delta$* flies (*Figure 3H–J*). They also exhibited increased *dome* expression (*Figure 3—figure supplement 1*). The effects of these manipulations on published *foxo* target genes were mixed (*Figure 3—figure supplement 1*); the strongest effect we observed was that Dome blockade increased *upd2* expression (*Figure 3—figure supplement 1*), consistent with the observation that FOXO activity inhibits *upd2* expression in muscle (none of the other genes tested have been shown to be FOXO targets in muscle) (*Zhao and Karpac, 2017*). This may explain some of the systemic effects of Dome blockade.

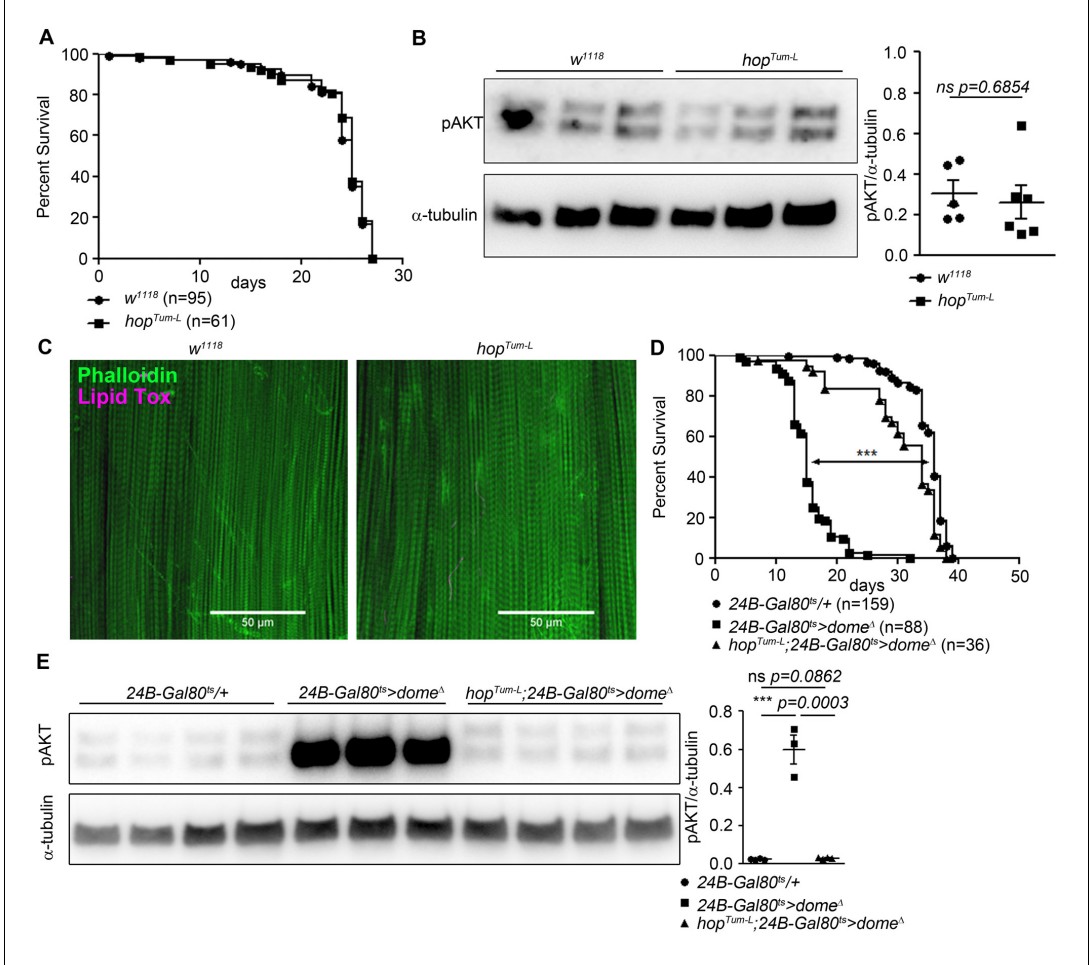

**Figure 2.** Hop is required, but not sufficient, for Dome to control AKT. (**A**) Lifespan of $w^{1118}$ and $hop^{Tum-L}$ flies at 29°C. Log-Rank test: $\chi^2 = 0.3223$, ns p=0.5702; Wilcoxon test: $\chi^2 = 0.4756$, ns p=0.4906. (**B**) Phospho-AKT in leg samples from 14-day-old $w^{1118}$ and $hop^{Tum-L}$ flies. One experiment of two is shown. Unpaired T-Test: ns p=0.6854. (**C**) Actin (Phalloidin) and neutral lipid (LipidTox) in flight muscle from 14-day-old $w^{1118}$ and $hop^{Tum-L}$ flies. One representative fly shown of six analysed per genotype. Scale bar = 50 μm. (**D**) Lifespan of $24B\text{-}Gal80^{ts}/+$, $24B\text{-}Gal80^{ts} > dome^{\Delta}$, and $hop^{Tum-L};24B\text{-}Gal80^{ts} > dome^{\Delta}$ flies at 29°C. Log-Rank test ($24B\text{-}Gal80^{ts}/+$ vs. $24B\text{-}Gal80^{ts} > dome^{\Delta}$): $\chi^2 = 319.4$, ***p<0.0001; Wilcoxon test ($24B\text{-}Gal80^{ts}/+$ vs. $24B\text{-}Gal80^{ts} > dome^{\Delta}$): $\chi^2 = 280.2$, ***p<0.0001. Log-Rank test ($24B\text{-}Gal80^{ts}/+$ vs. $hop^{Tum-L}$ $24B\text{-}Gal80^{ts} > dome^{\Delta}$): $\chi^2 = 18.87$, ***p<0.0001; Wilcoxon test ($24B\text{-}Gal80^{ts}/+$ vs. $hop^{Tum-L}$ $24B\text{-}Gal80^{ts} > dome^{\Delta}$): $\chi^2 = 20.83$, ***p<0.0001. (**E**) Phospho-AKT in leg samples from 14-day-old $24B\text{-}Gal80^{ts}/+$, $24B\text{-}Gal80^{ts} > dome^{\Delta}$ and $hop^{Tum-L};24B\text{-}Gal80^{ts} > dome^{\Delta}$ flies. P values from unpaired T-Test.

The online version of this article includes the following figure supplement(s) for figure 2:

**Figure supplement 1.** Interactions of *dome* with AMPK, MAPK, and FOXO signalling in adult muscle.

The effect of the *foxo* transgene was stronger than expected from a 1.5-fold increase in *foxo* expression, so we further explored the relationship between FOXO protein expression and AKT phosphorylation. We found that $24B\text{-}Gal80^{ts} > dome^{\Delta}$ markedly increased FOXO-GFP abundance, so that the increase in total FOXO was much greater than 1.5-fold (**Figure 3—figure supplement 1**). This drove an apparent feedback effect, restoring AKT in leg samples of $foxo^{GFP};24B\text{-}Gal80^{ts} > dome^{\Delta}$ flies to near-normal levels (**Figure 3—figure supplement 1**).

We also analysed expression of the catabolic hormone Akh and its putative targets *bmm*, *Hsl*, *plin1* and *plin2* in $24B\text{-}Gal80^{ts} > dome^{\Delta}$ animals (**Figure 3—figure supplement 1**). We observed no clear regulation of *Akh* itself or of *Hsl*, *bmm*, or *plin2*; *plin1* was reduced in expression by expression of *dome*$^{\Delta}$. We conclude that the primary effect of muscle *dome* is on AKT-foxo signalling.

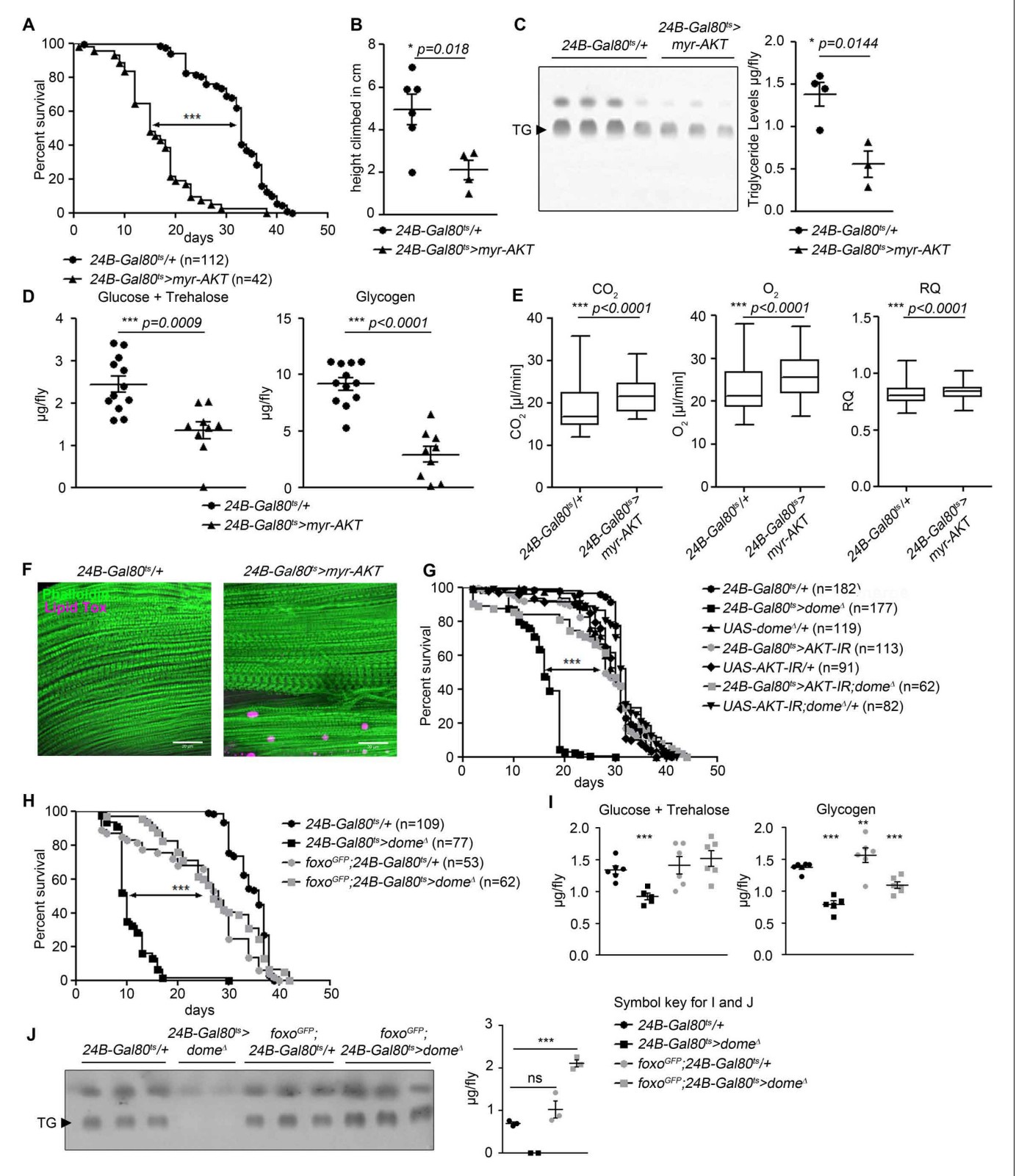

**Figure 3.** AKT hyperactivation causes pathology in *24B-Gal80ts > dome^Δ* flies. (**A**) Lifespan of *24B-Gal80ts/+* and *24B-Gal80ts > myr*-AKT at 29°C. Log-Rank test: $\chi^2$ = 115.5, ***p<0.0001; Wilcoxon test: $\chi^2$ = 123.6, ***p<0.0001. (**B**) Negative geotaxis assay of 14-day-old *24B-Gal80ts/+* and *24B-Gal80ts > myr*-AKT flies. Points represent mean height climbed in individual vials (~20 flies/vial), pooled from two independent experiments. Unpaired T-Test: *p=0.018. (**C**) TLC of triglycerides in 7-day-old *24B-Gal80ts/+* and *24B-Gal80ts > myr*-AKT flies, n = 3–4 per genotype. One experiment of two is

*Figure 3 continued on next page*

Figure 3 continued
shown. Unpaired T-Test: *p=0.0144. (D) Glucose and trehalose (left panel) and glycogen (right panel) in 7-day-old *24B-Gal80^ts/+* (n = 12) and *24B-Gal80^ts > myr*-AKT (n = 9) flies, pooled from two independent experiments. Unpaired T-Test (Glucose +Trehalose): ***p=0.0009 and unpaired T-Test (Glycogen): ***p<0.0001. (E) $CO_2$ produced, $O_2$ consumed, and RQ of 7-day-old *24B-Gal80^ts/+* and *24B-Gal80^ts > myr*-AKT flies. Box plots show data from one representative experiment of three, with data points collected from a 24 hr measurement pooled from 3 to 4 tubes per genotype with 10 flies/tube. P values from Mann-Whitney test. (F) Phalloidin and LipidTox staining of thorax samples from 14-day-old *24B-Gal80^ts/+* and *24B-Gal80^ts > myr*-AKT flies. One representative fly per genotype is shown of 3 analysed per group in two independent experiments. Scale bar = 50 μm. (G) Lifespan of *24B-Gal80^ts/+*, *24B-Gal80^ts > dome^Δ*, *UAS-dome^Δ/+*, *24B-Gal80^ts > AKT*-IR, *UAS-AKT-IR/+*, *24B-Gal80^ts > AKT-IR;dome^Δ* and *UAS-AKT-IR; dome^Δ/+* flies at 29°C. Log-Rank test (*24B-Gal80^ts > dome^Δ* vs. *24B-Gal80^ts > AKT-IR;dome^Δ*): $\chi^2$ = 101.0, ***p<0.0001; Wilcoxon test (*24B-Gal80^ts > dome^Δ* vs. *24B-Gal80^ts > AKT-IR;dome^Δ*): $\chi^2$ = 59.87, ***p<0.0001. (H) Lifespan of *24B-Gal80^ts/+*, *24B-Gal80^ts > dome^Δ*, *foxo-GFP;24B-Gal80^ts/+*, and *foxo-GFP;24B-Gal80^ts > dome^Δ* flies at 29°C. Log-Rank test (*24B-Gal80^ts > dome^Δ* vs. *foxo-GFP;24B-Gal80^ts > dome^Δ*): $\chi^2$ = 114.0, ***p<0.0001; Wilcoxon test (*24B-Gal80^ts > dome^Δ* vs. *foxo-GFP;24B-Gal80^ts > dome^Δ*): $\chi^2$ = 93.59, ***p<0.0001. (I) Glucose + trehalose and glycogen in 7-day-old *24B-Gal80^ts/+*, *24B-Gal80^ts > dome^Δ*, *foxo-GFP;24B-Gal80/+*, and *foxo-GFP; 24B-Gal80^ts > dome^Δ* flies. Statistical testing was performed with one-way ANOVA. (J) TLC of triglycerides in 7-day-old *24B-Gal80^ts/+*, *24B-Gal80^ts > dome^Δ*, *foxo-GFP;24B-Gal80^ts/+*, and *foxo-GFP;24B-Gal80^ts > dome^Δ* flies. Statistical testing was performed with one-way ANOVA.

The online version of this article includes the following figure supplement(s) for figure 3:

**Figure supplement 1.** Mutual regulation by AKT, Foxo, and Dome.

## Plasmatocytes are a relevant source of *upd* signals

Plasmatocytes—*Drosophila* macrophages—are a key source of *upd3* in flies on high fat diet and in mycobacterial infection (*Péan et al., 2017*; *Woodcock et al., 2015*). Plasmatocytes also express *upd1-3* in unchallenged flies (*Chakrabarti et al., 2016*). We thus tested their role in activation of muscle Dome.

We found plasmatocytes close to STAT-GFP-positive leg muscle (*Figure 4A,B*). This, and the prior published data, suggested that plasmatocytes might produce relevant levels of *dome*-activating cytokines in steady state. We then overexpressed *upd3* in plasmatocytes and observed a potent increase in muscle STAT-GFP activity (*Figure 4C*), confirming that plasmatocyte-derived *upd* signals were able to activate muscle Dome.

To determine the physiological relevance of plasmatocyte-derived signals, we assayed STAT-GFP activity in flies in which plasmatocytes had been depleted by expression of the pro-apoptotic gene *reaper* (*rpr*) using a temperature-inducible plasmatocyte-specific driver line (*w;tub-Gal80^ts;crq-Gal4*). These animals exhibited a near-complete elimination of plasmatocytes within 24 hr of being shifted to 29°C (*Figure 4—figure supplement 1*). STAT-GFP fluorescence and GFP abundance were reduced in legs of plasmatocyte-depleted flies (*crq-Gal80^ts > rpr*) compared to controls (*crq-Gal80^ts/+*) (*Figure 4D,E*). Activity was not eliminated, indicating that plasmatocytes are not the only source of muscle STAT-activating signals, although these animals did exhibit a significant reduction in climbing activity (*Figure 4—figure supplement 1*).

We then examined the lifespan of flies in which we had depleted plasmatocytes in combination with various *upd* mutations and knockdowns. Plasmatocyte depletion gave animals that were short-lived (*Figure 4F*). (This effect was different from that we previously reported, possibly due to changes in fly culture associated with an intervening laboratory move [*Woodcock et al., 2015*]). Combining plasmatocyte depletion with null mutations in *upd2* and *upd3* did not significantly further reduce lifespan; *upd2 upd3* mutants with plasmatocytes intact exhibited near-normal lifespan (*Figure 4F*). Similarly, plasmatocyte depletion drove muscle lipid accumulation, and *upd2 upd3* mutation synergised with plasmatocyte depletion to further increase muscle lipid inclusions (*Figure 4G*). Plasmatocyte depletion reduced free sugar levels as well as glycogen levels in the whole fly (*Figure 4—figure supplement 2*), but did not reduce the abundance of stored triglycerides (*Figure 4—figure supplement 2*). However, depleting plasmatocytes in *upd2 upd3* mutants failed to recapitulate the effects of muscle Dome inhibition on whole-animal triglyceride, free sugar, and glycogen levels (*Figure 4—figure supplement 2*). This could be due to antagonistic effects of other plasmatocyte-derived signals.

We attempted to pinpoint a specific Upd as the relevant physiological ligand by examining STAT-GFP activity, first testing mutants in *upd2* and *upd3* because *upd1* mutation is lethal. However, these mutants, including the *upd2 upd3* double-mutant, were apparently normal (*Figure 4—figure supplement 2*). We then tested plasmatocyte-specific knockdown of *upd1* and *upd3*; these animals

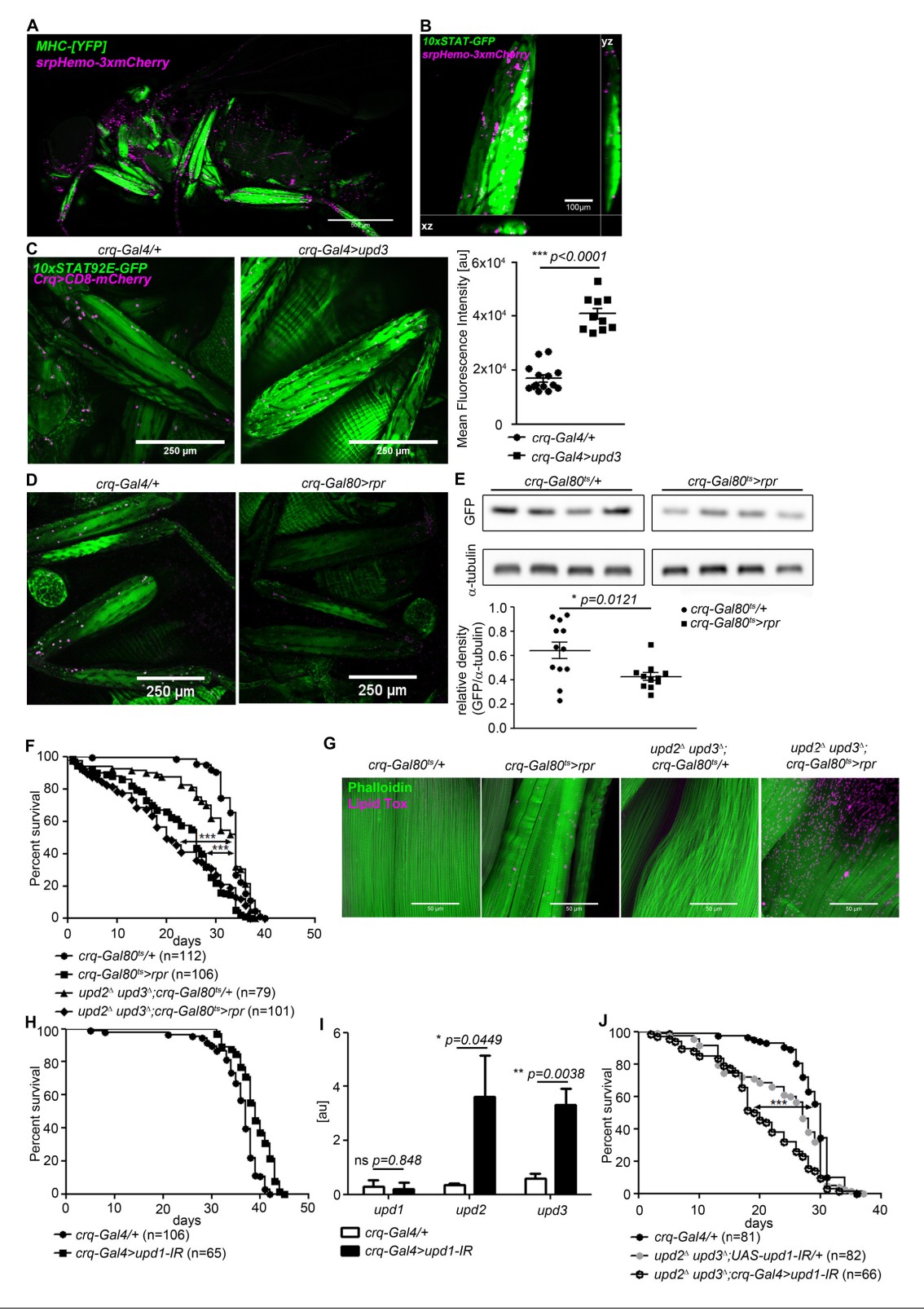

**Figure 4.** Plasmatocytes promote muscle Dome activity. (**A**) Muscle (*MHC^YFP*) and plasmatocytes (*srpHemo-3xmCherry*) in 7-day-old flies. Plasmatocytes are found in close proximity to adult muscles. One representative fly of 5 is shown. Scale bar = 500 μm. (**B**) Legs and plasmatocytes in 7-day-old *10xSTAT92E-GFP;srpHemo-3xmCherry* flies. Muscle with high JAK-STAT activity (green) is surrounded by plasmatocytes (magenta). One representative fly of 5 is shown. Scale bar = 100 μm. (**C**) STAT activity and plasmatocytes in legs from control (*10xSTAT92E-GFP;crq-Gal4 >CD8-mCherry/+*) and *upd3-*
*Figure 4 continued on next page*

*Figure 4 continued*

overexpressing (*10xSTAT92E-GFP;crq-4>CD8mCherry/UAS-upd3*) flies. One representative fly of 10–14 is shown. Scale bar = 100 µm. Graph shows mean fluorescence intensity (MFI). Unpaired T-Test: ***p<0.0001. (D) STAT activity and plasmatocytes in legs from control (*10xSTAT92E-GFP;crq-Gal80$^{ts}$ > CD8-mCherry/+*) and plasmatocyte-depleted (*10xSTAT92E-GFP;crq-Gal80$^{ts}$ > CD8 mCherry/rpr*) flies. One representative fly of six is shown. Scale bar = 250 µm. (E) Western blot analysis of STAT-driven GFP in legs from 7-day-old control (*10xSTAT92E-GFP;crq-Gal80$^{ts}$ > CD8-mCherry/+*) and plasmatocyte-depleted (*10xSTAT92E-GFP;crq-Gal80$^{ts}$ > CD8 mCherry/rpr* flies). One representative experiment of three is shown. Graph shows STAT-GFP/α-tubulin for control (*crq-Gal80$^{ts}$/+*) and plasmatocyte-depleted (*crq-Gal80$^{ts}$ > rpr*) leg samples. Unpaired T-Test: *p=0.0121. (F) Lifespan of *crq-Gal80$^{ts}$/+*, *crq-Gal80$^{ts}$ > rpr*, *upd2$^\Delta$ upd3$^\Delta$;crq-Gal80$^{ts}$/+*, and *upd2$^\Delta$ upd3$^\Delta$;crq-Gal80$^{ts}$ > rpr* flies at 29°C. Log-Rank test (*crq-Gal80$^{ts}$/+* vs. *crq-Gal80$^{ts}$ > rpr*): χ2 = 101.7, ***p<0.0001; Wilcoxon test (*crq-Gal80$^{ts}$/+* vs. *crq-Gal80$^{ts}$ > rpr*): χ2 = 107.8, ***p<0.0001; Log-Rank test (*crq-Gal80$^{ts}$/+* vs. *upd2$^\Delta$ upd3$^\Delta$;crq-Gal80$^{ts}$ > rpr*): χ2 = 60.03, ***p<0.0001; Wilcoxon test (*crq-Gal80$^{ts}$/+* vs. *upd2$^\Delta$ upd3$^\Delta$;crq-Gal80$^{ts}$ > rpr*): χ2 = 80.97, ***p<0.0001. (G) Actin (Phalloidin) and neutral lipid (LipidTox) in thorax samples from 14-day-old *crq-Gal80$^{ts}$/+*, *crq-Gal80$^{ts}$ > rpr*, *upd2$^\Delta$ upd3$^\Delta$;crq-Gal80$^{ts}$/+*, and *upd2$^\Delta$ upd3$^\Delta$;crq-Gal80$^{ts}$ > rpr* flies. One representative fly per genotype shown of 6 analysed per group. Scale bar = 50 µm. (H) Lifespan of *crq-Gal4/+* and *crq-Gal4 >upd1* IR flies at 29°C. Log-Rank test: χ2 = 31.36, ***p<0.0001; Wilcoxon test: χ2 = 22.17, ***p=0.0001. (I) Expression by qRT-PCR of *upd1*, *upd2* and *upd3* in thorax samples of *crq-Gal4/+* and *crq-Gal4 >upd1* IR flies, data from four independent samples of each genotype.. Unpaired T-Test (*upd1*): ns p=0.848, unpaired T-Test (*upd2*): *p=0.0449 and unpaired T-Test (*upd3*): **p=0.0038. (J) Lifespan of *crq-Gal4/+*, *upd2$^\Delta$ upd3$^\Delta$;UAS-upd1-IR/+*, and *upd2$^\Delta$ upd3$^\Delta$;crq-Gal4 >upd1* IR flies at 29°C. Log-Rank test (*crq-Gal4/+* vs. *upd2$^\Delta$ upd3$^\Delta$;crq-Gal4 >upd1* IR): χ2 = 41.12, ***p<0.0001; Wilcoxon test (*crq-Gal4/+* vs. *upd2$^\Delta$ upd3$^\Delta$;crq-Gal4 >upd1* IR): χ2 = 54.47, ***p<0.0001 Log-Rank test (*crq-Gal4/+* vs. *upd2$^\Delta$ upd3$^\Delta$;UAS-upd1-IR/+*): χ2 = 14.46, ***p<0.0001; Wilcoxon test (*crq-Gal4/+* vs. *upd2$^\Delta$ upd3$^\Delta$;UAS-upd1-IR/+*): χ2 = 19.99, ***p<0.0001. P values in C, E, H from unpaired T-test.

The online version of this article includes the following figure supplement(s) for figure 4:

**Figure supplement 1.** Further characterisation of plasmatocyte-depleted flies.
**Figure supplement 2.** Further characterisation of requirements for specific Upds.

were also essentially normal (*Figure 4—figure supplement 2*), and plasmatocyte *upd1* knockdown did not reduce lifespan (*Figure 4H*). However, plasmatocyte-specific *upd1* knockdown gave significant compensating increases in expression of *upd2* and *upd3* (*Figure 4I*). In keeping with this, combining plasmatocyte-specific *upd1* knockdown with mutations in *upd2* and *upd3* reduced lifespan (*Figure 4J*) and also reduced STAT-GFP activity in these flies (*Figure 4—figure supplement 2*).

Our results indicate that plasmatocytes are an important physiological source of the Upd signal driving muscle Dome activity in healthy flies, and suggest that *upd1* may be the primary relevant signal in healthy animals. However, plasmatocytes are not the only relevant source of signal, and Upd mutual regulation prevents us from pinpointing a single responsible signal.

## Discussion

Here we show that *upd-dome* signalling in muscle acts via AKT to regulate physiological homeostasis in *Drosophila*. Loss of Dome activity in adult muscles shortens lifespan and promotes local and systemic metabolic disruption. Dome specifically regulates the level and activity of AKT; AKT hyper-activation mediates the observed pathology. Plasmatocytes are a primary source of the cytokine signal. In healthy adult flies, insulin-like peptides are the primary physiological AKT agonists. The effect we observe thus appears to be an example of a cytokine-Dome-JAK signal that impairs insulin function to permit healthy physiology.

Our work fits into a recent body of literature demonstrating key physiological roles for JAK-STAT activating signals in *Drosophila*. Upd1 acts locally in the brain to regulate feeding and energy storage by altering the secretion of neuropeptide F (NPF) (*Beshel et al., 2017*). Upd2 is released by the fat body in response to dietary triglyceride and sugar to regulate secretion of insulin-like peptides (*Rajan and Perrimon, 2012*). More recently, muscle-derived Upd2, under control of FOXO, has been shown to regulate production of the glucagon-like signal Akh (*Zhao and Karpac, 2017*). Indeed, we observe that *upd2* is upregulated in flies with Dome signalling blocked in muscle, possibly explaining some of the systemic metabolic effects we observe. Plasmatocyte-derived Upd3 in flies on a high fat diet can activate the JAK/STAT pathway in various organs including muscles and can promote insulin insensitivity (*Woodcock et al., 2015*). Our observation that Upd signalling is required to control AKT accumulation and thus insulin pathway activity in healthy adult muscle may explain some of these prior observations and reveals a new role for plasmatocyte-derived cytokine signalling in healthy metabolic regulation.

Several recent reports have examined roles of JAK/STAT signalling in *Drosophila* muscle. In larvae, muscle JAK/STAT signalling can have an effect opposite to the one we report, with pathway

loss of function resulting in reduced AKT activity (*Yang and Hultmark, 2017*). It is unclear whether this difference represents a difference in function between developmental stages (larva vs adult) or a difference between acute and chronic consequences of pathway inactivation. Roles in specific muscle populations have also been described: for example, JAK/STAT signalling in adult visceral muscle regulates expression of Vein, an EGF-family ligand, to control intestinal stem cell proliferation (*Buchon et al., 2010*; *Jiang et al., 2011*); the role of this system in other muscles may be analogous, controlling expression of various signals to regulate systemic physiology. Importantly, though we do not observe loss of gut integrity in our flies, it remains possible that the gut is an important mediator of some aspect of the physiological *unpaired* signal we document—either acting as an endocrine relay or via more subtle effects on gut physiology that could affect nutrient absorption. This would fit our observation that expression of dominant-negative *Dome* under the control of Mhc-Gal4 (which does not express in visceral muscle) gives a weaker effect on survival and AKT abundance than other muscle drivers and is particularly of interest given the documented role of plasmatocytes in regulation of gut homeostasis (*Ayyaz et al., 2015*).

The roles of mammalian JAK/STAT signalling in muscle physiology are more complex, but exhibit several parallels with the fly. In mice, early muscle-specific deletion of Growth Hormone Receptor (GHR) causes several symptoms including insulin resistance, while adult muscle-specific GHR deletion causes entirely different effects, including increased metabolic rate and insulin sensitivity on a high-fat diet (*Mavalli et al., 2010*; *Vijayakumar et al., 2013*; *Vijayakumar et al., 2012*). GHR signals via STAT5; STAT5 deletion in adult skeletal muscle promotes muscle lipid accumulation on a high-fat diet (*Baik et al., 2017*). Other STAT pathways can also play roles. For example, the JAK-STAT activating cytokine IL-6, which signals primarily via STAT3, increases skeletal muscle insulin sensitivity when given acutely but can drive insulin resistance when provided chronically (*Nieto-Vazquez et al., 2008*). STAT3 itself can promote muscle insulin resistance (*Kim et al., 2013*; *Mashili et al., 2013*). The relationship between these effects and those we have shown here, and the mechanisms regulating Upd production by plasmatocytes during healthy physiology, remain to be determined.

# Materials and methods

## Key resources table

| Reagent type (species) or resource | Designation | Source or reference | Identifiers | Additional information |
|---|---|---|---|---|
| Genetic reagent (*D. melanogaster*) | $w^{1118}$; tubulin-Gal80$^{ts}$/ SM6a;24B-Gal4/TM6c, Sb$^1$ | This study | | Inserted Elements: P[w[+mC]=tub P-GAL80[ts]]; P[GawB]how24B |
| Genetic reagent (*D. melanogaster*) | $w^{1118}$; tubulin-Gal80$^{ts}$/ SM6a;Mef2-Gal4/TM6c, Sb$^1$ | This study | | Inserted Elements: P[w[+mC]=tub P-GAL80[ts]]; P[GAL4-Mef2.R]3 |
| Genetic reagent (*D. melanogaster*) | $w^{1118}$;;UAS-dome$^\Delta$/TM6c, Sb$^1$ | *Brown et al., 2001* | | Gift of James Castelli-Gair Hombría |
| Genetic reagent (*D. melanogaster*) | $w^{1118}$;UAS-dome$^\Delta$/CyO | *Brown et al., 2001* | | Gift of James Castelli-Gair Hombría |
| Genetic reagent (*D. melanogaster*) | $w^{1118}$;;UAS-myr-AKT/TM6c, Sb$^1$ | *Stocker et al., 2002* | | Gift of Ernst Hafen |
| Genetic reagent (*D. melanogaster*) | w;UAS-AMPKα-IR | Vienna *Drosophila* Research Center (VDRC) | RRID: FlyBase_FBst0478025; VDRC 106200 | |
| Genetic reagent (*D. melanogaster*) | w;UAS-AMPKβ-IR | VDRC | RRID: FlyBase_FBst0476347; VDRC 104489 | |
| Genetic reagent (*D. melanogaster*) | w;UAS-rl-IR | VDRC | RRID: FlyBase_FBst0480887; VDRC 109108 | |

*Continued on next page*

Continued

| Reagent type (species) or resource | Designation | Source or reference | Identifiers | Additional information |
|---|---|---|---|---|
| Genetic reagent (D. melanogaster) | w;UAS-Dsor1-IR | VDRC | RRID: FlyBase_FBst0479098; VDRC 107276 | |
| Genetic reagent (D. melanogaster) | w[1118];foxo[GFP] | BDSC | RRID:BDSC_38644 | Inserted Element: PBac[foxo-GFP. FLAG]VK00037 |
| Genetic reagent (D. melanogaster) | w;UAS-AKT-IR | VDRC | RRID: FlyBase_FBst0475561; VDRC 103703 | |
| Genetic reagent (D. melanogaster) | w[1118];10xSTAT92E-GFP | BDSC *Bach et al., 2007* | RRID:BDSC_26197 | Inserted Element: P[10XStat92E-GFP]1 |
| Genetic reagent (D. melanogaster) | w[1118];MHC-Gal4, MHC-RFP/SM6a | BDSC | RRID:BDSC_38464 | Inserted Element: P[Mhc-GAL4.F3-580]2; P[Mhc-RFP.F3-580]2 |
| Genetic reagent (D. melanogaster) | w upd2[Δ] upd3[Δ];;; | BDSC | RRID:BDSC_55729 | |
| Genetic reagent (D. melanogaster) | w[1118];;crq-Gal4/TM6 c, Sb[1] | | | Gift of Nathalie Franc |
| Genetic reagent (D. melanogaster) | w[1118];tub-Gal80[ts];TM2/TM6 c, Sb[1] | BDSC | RRID:BDSC_7108 | |
| Genetic reagent (D. melanogaster) | w[1118];;UAS-rpr/TM6 c, Sb[1] | BDSC | RRID:BDSC_5824 | |
| Genetic reagent (D. melanogaster) | w[1118];UAS-CD8-mCherry | BDSC | RRID:BDSC_27391 | |
| Genetic reagent (D. melanogaster) | w[1118];;srpHemo-3xmCherry/TM6c, Sb[1] | *Gyoergy et al., 2018* | | |
| Genetic reagent (D. melanogaster) | w;UAS-hop-IR | VDRC | RRID: FlyBase_FBst0463355; VDRC 40037 | |
| Genetic reagent (D. melanogaster) | w;UAS-upd1-IR/SM6a | VDRC | RRID: FlyBase_FBst0459787; VDRC 3282 | |
| Genetic reagent (D. melanogaster) | w;UAS-upd3-IR | VDRC | RRID: FlyBase_FBst0456774; VDRC 27134 | |
| Genetic reagent (D. melanogaster) | w[1118];;UAS-upd3/TM6 c, Sb[1] | | | Gift of Bruce Edgar |
| Genetic reagent (D. melanogaster) | w[1118];UAS-2xeGFP/SM6a | BDSC | RRID:BDSC_6874 | |
| Genetic reagent (D. melanogaster) | w[1118] hop[Tum-L]/FM7h | BDSC | RRID:BDSC_8492 | backcrossed onto w[1118] background |
| Sequence-based reagent | Akt1_forward | This study | PCR primers | 5'-ctttgcgagtattaactggacaga-3' |
| Sequence-based reagent | Akt1_reverse | This study | PCR primers | 5'-ggatgtcacctgaggcttg-3' |
| Sequence-based reagent | Ilp2_forward | This study | PCR primers | 5'-atcccgtgattccaccacaag-3' |
| Sequence-based reagent | Ilp2_reverse | This study | PCR primers | 5'-gcggttccgatatcgagtta-3' |
| Sequence-based reagent | Ilp3_forward | This study | PCR primers | 5'-caacgcaatgaccaagagaa-3' |
| Sequence-based reagent | Ilp3_reverse | This study | PCR primers | 5'-tgagcatctgaaccgaact-3' |

*Continued*

| Reagent type (species) or resource | Designation | Source or reference | Identifiers | Additional information |
|---|---|---|---|---|
| Sequence-based reagent | Ilp4_forward | This paper | PCR primers | 5'-gagcctgattagactgggactg-3' |
| Sequence-based reagent | Ilp4_reverse | This paper | PCR primers | 5'-tggaccggctgcagtaac-3' |
| Sequence-based reagent | Ilp5_forward | This paper | PCR primers | 5'-gccttgatggacatgctga-3' |
| Sequence-based reagent | Ilp5_reverse | This paper | PCR primers | 5'-agctatccaaatccgcca-3' |
| Sequence-based reagent | Ilp6_forward | This paper | PCR primers | 5'-cccttggcgatgtatttcc-3' |
| Sequence-based reagent | Ilp6_reverse | This paper | PCR primers | 5'-cacaaatcggttacgttctgc-3' |
| Sequence-based reagent | Ilp7_forward | This paper | PCR primers | 5'-cacaccgaggagggtctc-3' |
| Sequence-based reagent | Ilp7_reverse | This paper | PCR primers | 5'-caatatagctggcggacca-3' |
| Sequence-based reagent | dome_forward | This paper | PCR primers | 5'-cggactttcggtactccatc-3' |
| Sequence-based reagent | dome_reverse | This paper | PCR primers | 5'-accttgatgaggccaggat-3' |
| Sequence-based reagent | upd1_forward | This paper | PCR primers | 5'-gcacactgatttcgatacgg-3' |
| Sequence-based reagent | upd1_reverse | This paper | PCR primers | 5'- ctgccgtggtgctgtttt —3' |
| Sequence-based reagent | upd2_forward | This paper | PCR primers | 5'-cggaacatcacgatgagcgaat-3' |
| Sequence-based reagent | upd2_reverse | This paper | PCR primers | 5'-tcggcaggaacttgtactcg-3' |
| Sequence-based reagent | upd3_forward | This paper | PCR primers | 5'-actgggagaacacctgcaat-3' |
| Sequence-based reagent | upd3_reverse | This paper | PCR primers | 5'-gcccgtttggttctgtagat-3' |
| Sequence-based reagent | Pepck_forward | This paper | PCR primers | 5'-ggataaggtggacgtgaag-3' |
| Sequence-based reagent | Pepck_reverse | This paper | PCR primers | 5'-acctcctgcgaccagaact-3' |
| Sequence-based reagent | Thor_forward | This paper | PCR primers | 5'-caggaaggttgtcatctcgga-3' |
| Sequence-based reagent | Thor_reverse | This paper | PCR primers | 5'-ggagtggtggagtagagggtt-3' |
| Sequence-based reagent | InR_forward | This paper | PCR primers | 5'-gcaccattataaccggaacc-3' |
| Sequence-based reagent | InR_reverse | This paper | PCR primers | 5'-ttaattcatccatgacgtgagc-3' |
| Sequence-based reagent | Akh_forward | This paper | PCR primers | 5'- agccgtgctcttcatgct-3' |
| Sequence-based reagent | Akh_reverse | This paper | PCR primers | 5'-aaaggttccaggaccagctc-3' |
| Sequence-based reagent | Hsl_forward | This paper | PCR primers | 5'-cttggaaatacttgaggggttg-3' |
| Sequence-based reagent | Hsl_reverse | This paper | PCR primers | 5'-agatttgatgcagttctttgagc-3' |

*Continued on next page*

*Continued*

| Reagent type (species) or resource | Designation | Source or reference | Identifiers | Additional information |
|---|---|---|---|---|
| Sequence-based reagent | bmm_forward | This paper | PCR primers | 5'-gtctcctctgcgatttgccat-3' |
| Sequence-based reagent | bmm_reverse | This paper | PCR primers | 5'-ctgaagggacccagggagta-3' |
| Sequence-based reagent | plin1_forward | This paper | PCR primers | 5'-gcgttctatggtagccttcag-3' |
| Sequence-based reagent | plin1_reverse | This paper | PCR primers | 5'-gcgtccggatagaaagctg-3' |
| Sequence-based reagent | plin2_forward | This paper | PCR primers | 5'-gcagaatggcaagagttctga-3' |
| Sequence-based reagent | plin2_reverse | This paper | PCR primers | 5'-actgtgtgtaggactggatcctc-3' |
| Sequence-based reagent | Rpl1_forward | This paper | PCR primers | 5'-tccaccttgaagaagggcta-3' |
| Sequence-based reagent | Rpl1_reverse | This paper | PCR primers | 5'-ttgcggatctcctcagactt-3' |
| Peptide, recombinant protein | Trehalase | Sigma Aldrich | T8778 | |
| Peptide, recombinant protein | β-Amyloglucosidase | Sigma Aldrich | 10115 | |
| Antibody | anti-phospho (Ser505)-AKT | Cell Signal Technology (CST) | Cat# 4054; RRID:AB_331414 | WB (1:1000) |
| Antibody | anti-AKT | Cell Signal Technology (CST) | Cat# 4691; RRID:AB_915783 | WB (1:1000) |
| Antibody | anti-phospho (Thr172)-AMPKα | Cell Signal Technology (CST) | Cat# 2535; RRID:AB_331250 | WB (1:1000) |
| Antibody | anti-phospho (Thr389)-p70 S6 kinase | Cell Signal Technology (CST) | Cat# 9206; RRID:AB_2285392 | WB (1:1000) |
| Antibody | anti-GFP | Cell Signal Technology (CST) | Cat# 2956; RRID:AB_1196615 | WB (1:1000) |
| Antibody | anti-phospho-p44 /42 MAPK (Erk1/2) | Cell Signal Technology (CST) | Cat# 4370; RRID:AB_2315112 | WB (1:1000) |
| Antibody | anti-α-tubulin | Developmental Studies Hybridoma Bank) | Clone 12G10; RRID:AB_1157911 | WB (1:3000) |
| Antibody | HRP anti-rabbit IgG | Cell Signal Technology (CST) | Cat# 7074; RRID:AB_2099233 | WB (1:5000) |
| Antibody | HRP anti-mouse IgG | Cell Signal Technology (CST) | Cat# 7076; RRID:AB_330924 | WB (1:5000) |
| Commercial assay or kit | First Strand cDNA Synthesis Kit | Thermo Scientific | K1622 | |
| Commercial assay or kit | Sensimix SYBR Green no-ROX | Bioline | QT650-05 | |
| Chemical compound, drug | Bromophenol blue | Sigma Aldrich | SML1656 | |
| Chemical compound, drug | Xylene cyanol | Carl Roth | A513.1 | |
| Chemical compound, drug | Brilliant Blue FCF | Sigma Aldrich | 80717 | |
| Other | HCS Lipid Tox Deep Red | Thermo Fisher | H34477 | IF (1:200) |
| Other | Alexa Fluor 488 Phalloidin | Thermo Fisher | A12379 | IF (1:20) |
| Other | Fluoromount-G | ebioscience | 00-4958-02 | |

*Continued on next page*

Continued

| Reagent type (species) or resource | Designation | Source or reference | Identifiers | Additional information |
|---|---|---|---|---|
| Other | TRIzol | Invitrogen | AM9738 | |
| Other | Fixable Viability Dye 780 | ebioscience | 65-0865-18 | FC (1:1000) |
| Other | Supersignal West Pico Chemilu minescent Substrate | Thermo Scientific | 34077 | |
| Other | Supersignal West Femto Chemiluminescent Substrate | Thermo Scientific | 34094 | |
| Other | Glucose Reagent | Sentinel Diagnostics | 17630H | |
| Software, algorithm | ImageJ | ImageJ | RRID:SCR_003070 | |
| Software, algorithm | GraphPad Prism | GraphPad | RRID:SCR_002798 | |
| Software, algorithm | FlowJo | FlowJo | RRID:SCR_008520) | |

## *Drosophila melanogaster* stocks and culture

All fly stocks were maintained on food containing 10% w/v Brewer's yeast, 8% fructose, 2% polenta and 0.8% agar supplemented with propionic acid and nipagin. Crosses for experiments were performed at 18°C (for crosses with temperature inducible gene expression) or 25°C. Flies were shifted to 29°C after eclosion where relevant.

Male flies were used for all experiments. All flies were backcrossed onto our laboratory isogenic $w^{1118}$ genetic background, with the exception of VDRC knockdown lines (these lines are also on a uniform genetic background and could be compared with one another). All crosses were performed using driver females so that the male progeny used for experiments would have a uniform X chromosome.

The following original fly stocks were used for crosses:

| Fly stocks | Description and origin |
|---|---|
| $w^{1118}$; tubulin-Gal80$^{ts}$/SM6a;24B-Gal4/TM6c, Sb$^1$ | Temperature sensitive muscle specific driver line; 24B-Gal4 a gift of Nazif Alic |
| $w^{1118}$; tubulin-Gal80$^{ts}$/SM6a;Mef2-Gal4/TM6c, Sb$^1$ | Temperature sensitive muscle specific driver line; Mef2-Gal4 a gift of Michael Taylor |
| $w^{1118}$;;UAS-dome$^\Delta$/TM6c, Sb$^1$ | Line for expression of a dominant-negative *dome*, gift of James Castelli-Gair Hombría |
| $w^{1118}$;UAS-dome$^\Delta$/CyO | Line for expression of a dominant-negative *dome*, gift of James Castelli-Gair Hombría |
| $w^{1118}$;;UAS-myr-AKT/TM6c, Sb$^1$ | Line for over-expression of a constitutive active (myristoylated) AKT, gift of Ernst Hafen |
| w;UAS-AMPKα-IR | VDRC KK106200 |
| w;UAS-AMPKβ-IR | VDRC KK104489 |
| w;UAS-rl-IR | VDRC KK109108 |
| w;UAS-Dsor1-IR | VDRC KK102276 |
| $w^{1118}$;foxo$^{GFP}$ | Expresses GFP-tagged *foxo* fusion protein (genomic rescue construct inserted at AttP40). Bloomington *Drosophila* Stock Center (BDSC) 38644 |
| w;UAS-AKT-IR | VDRC KK103703 |

*Continued on next page*

Continued

| Fly stocks | Description and origin |
|---|---|
| $w^{1118}$;10xSTAT92E-GFP | STAT-GFP reporter line (*Bach et al., 2007*). BDSC #26197 |
| $w^{1118}$;MHC-Gal4,MHC-RFP/SM6a | Muscle specific driver line and muscle specific reporter line. Derived from BDSC #38464 |
| $w$ $upd2^{\Delta}$ $upd3^{\Delta}$;;; | Gift of Bruno Lemaitre |
| $w^{1118}$;;crq-Gal4/TM6c, $Sb^1$ | Plasmatocyte specific driver line, gift of Nathalie Franc |
| $w^{1118}$;tub-Gal80$^{ts}$;TM2/TM6c, $Sb^1$ | Line for ubiquitous expression of *Gal80$^{ts}$*, BDSC #7108 |
| $w^{1118}$;;UAS-rpr/TM6c, $Sb^1$ | Line for over-expression of the pro-apoptotic protein rpr. Derived from BDSC #5824 |
| $w^{1118}$;UAS-CD8-mCherry | Line for overexpression of a CD8-mCherry fusion protein. Derived from BDSC #27391 |
| $w^{1118}$;;srpHemo-3xmCherry/TM6c, $Sb^1$ | Plasmatocyte reporter line (*Gyoergy et al., 2018*) |
| w;UAS-hop-IR | VDRC GD40037 |
| w;UAS-upd1-IR/SM6a | VDRC GD3282 |
| w;UAS-upd3-IR | VDRC GD6811 |
| $w^{1118}$;;UAS-upd3/TM6c, $Sb^1$ | Line for overexpression of upd3, gift of Bruce Edgar |
| $w^{1118}$;UAS-2xeGFP/SM6a | Line for expression of bicistronic GFP, BDSC #6874 |
| $w^{1118}$ $hop^{Tum-L}$/FM7h | Gain-of function mutant of *hop*; derived by backcrossing from BDSC 8492 onto our control $w^{1118}$ background |

Genotype abbreviations were used for the different experimental flies in this study, in the following table the complete genotypes are indicated:

| Genotype abbreviation of flies used in the manuscript | Complete genotype of flies used in the manuscript |
|---|---|
| 10XSTAT92E-GFP/MHC-RFP | $w^{1118}$;10xSTAT92E-GFP/MHC-Gal4,MHC-RFP |
| 24B-Gal80$^{ts}$/+ | $w^{1118}$;tub-Gal80$^{ts}$/+;24B-Gal4/+ |
| 24B-Gal80$^{ts}$ > dome$^{\Delta}$ | $w^{1118}$;tub-Gal80$^{ts}$/+;24B-Gal4/UAS-dome$^{\Delta}$ |
| 24B-Gal80$^{ts}$ > myr-AKT | $w^{1118}$;tub-Gal80$^{ts}$/+;24B-Gal4/UAS-myr-AKT |
| 24B-Gal80$^{ts}$ > AMPKα-IR | $w^{1118}$;tub-Gal80$^{ts}$/UAS-AMPKα-IR;24B-Gal4/+ |
| 24B-Gal80$^{ts}$ > AMPKβ-IR | $w^{1118}$;tub-Gal80$^{ts}$/UAS-AMPKβ-IR;24B-Gal4/+ |
| 24B-Gal80$^{ts}$ > rl-IR | $w^{1118}$;tub-Gal80$^{ts}$/UAS-rl-IR;24B-Gal4/+ |
| 24B-Gal80$^{ts}$ > Dsor1-IR | $w^{1118}$;tub-Gal80$^{ts}$/UAS-Dsor1-IR;24B-Gal4/+ |
| 24B-Gal80 > hop-IR | $w^{1118}$;tub-Gal80$^{ts}$/UAS-hop-IR;24B-Gal4/+ |
| hop$^{tum-L}$;24B-Gal80 > dome$^{\Delta}$ | $w^{1118}$ hop$^{tum-L}$;tub-Gal80$^{ts}$/+;24B-Gal4/UAS-dome$^{\Delta}$ |
| 24B-Gal80$^{ts}$ > AKT-IR | $w^{1118}$;tub-Gal80$^{ts}$/UAS-AKT-IR;24B-Gal4/+ |
| 24B-Gal80$^{ts}$ > AKT-IR;dome$^{\Delta}$ | $w^{1118}$;tub-Gal80$^{ts}$/UAS-AKT-IR;24B-Gal4/UAS-dome$^{\Delta}$ |
| MHC-Gal4/+ | $w^{1118}$;MHC-Gal4,Mhc-RFP/+; |
| MHC-Gal4 > dome$^{\Delta}$ (II) | $w^{1118}$;MHC-Gal4,MHC-RFP/UAS-dome$^{\Delta}$; |
| foxo$^{GFP}$;24B-Gal80$^{ts}$/+ | $w^{1118}$;foxo$^{GFP}$;tub-Gal80$^{ts}$/+;24B-Gal4/+ |
| foxo$^{GFP}$;24B-Gal80$^{ts}$ > dome$^{\Delta}$ | $w^{1118}$;foxo$^{GFP}$;tub-Gal80$^{ts}$/+;24B-Gal4/UAS-dome$^{\Delta}$ |
| UAS-dome$^{\Delta}$/+ | $w^{1118}$;;UAS-dome$^{\Delta}$/+ |
| UAS-AKT-IR/+ | $w^{1118}$;UAS-AKT-IR/+; |
| UAS-AKT-IR;dome$^{\Delta}$/+ | $w^{1118}$;UAS-AKT-IR/+; UAS-dome$^{\Delta}$/+ |
| Mef2-Gal80$^{ts}$/+ | $w^{1118}$;tub-Gal80$^{ts}$/+;Mef2-Gal4/+ |
| Mef2-Gal80$^{ts}$ > dome$^{\Delta}$ | $w^{1118}$;tub-Gal80$^{ts}$/+;Mef2-Gal4/UAS-dome$^{\Delta}$ |
| srpHemo-3xmCherry | $w^{1118}$;; srpHemo-3xmCherry/+ |

*Continued*

| Genotype abbreviation of flies used in the manuscript | Complete genotype of flies used in the manuscript |
|---|---|
| crq-Gal4/+ | $w^{1118}$;;crq-Gal4/+ |
| crq-Gal80$^{ts}$ > rpr | $w^{1118}$;tub-Gal80$^{ts}$/+;crq-Gal4/UAS-rpr or $w^{1118}$;tub-Gal80$^{ts}$/+;crq-Gal4,UAS-CD8-mCherry,10xSTAT92E-GFP/UAS-rpr |
| crq-Gal80$^{ts}$/+ | $w^{1118}$;tub-Gal80$^{ts}$/+;crq-Gal4/+ or $w^{1118}$;tub-Gal80$^{ts}$/+; crq-Gal4,UAS-CD8-mCherry,10xSTAT92E-GFP/+ |
| crq-Gal4/+ | $w^{1118}$;;crq-Gal4,UAS-CD8-mCherry,10xSTAT92E-GFP/+ |
| crq-Gal4 > upd1-IR | $w^{1118}$;UAS-upd1-IR/+;crq-Gal4,UAS-CD8-mCherry,10xSTAT92E-GFP/+ |
| crq-Gal4 > upd3-IR | $w^{1118}$;UAS-upd3-IR/+;crq-Gal4,UAS-CD8-mCherry,10xSTAT92E-GFP/+ |
| crq-Gal4 > upd3 | $w^{1118}$;;crq-Gal4,UAS-CD8-mCherry,10xSTAT92E-GFP/UAS-upd3 |
| upd2$^\Delta$ upd3$^\Delta$;crq-Gal80$^{ts}$/+ | w upd2$^\Delta$ upd3$^\Delta$;tub-Gal80$^{ts}$/+;crq-Gal4/+ |
| upd2$^\Delta$ upd3$^\Delta$;crq-Gal80$^{ts}$ > rpr | w upd2$^\Delta$ upd3$^\Delta$;tub-Gal80$^{ts}$/+;crq-Gal4/UAS-rpr |
| upd2$^\Delta$ upd3$^\Delta$;upd1-IR/+ | w upd2$^\Delta$ upd3$^\Delta$;UAS-upd1-IR/+ |
| upd2$^\Delta$ upd3$^\Delta$;crq-Gal4/+ | w upd2$^\Delta$ upd3$^\Delta$;;crq-Gal4/+ |
| upd2$^\Delta$ upd3$^\Delta$;crq-Gal4 > upd1-IR | w upd2$^\Delta$ upd3$^\Delta$;UAS-upd1-IR/+;crq-Gal4/+ |
| MHC$^{YFP}$; srpHemo-3xmCherry | $w^{1118}$; MHC$^{YFP}$/+;srpHemo-3xmCherry/+ |
| 10xSTAT92E-GFP; srpHemo-3xmCherry | $w^{1118}$; 10xSTAT92E-GFP/+;srpHemo-3xmCherry/+ |

## Lifespan/Survival assays

Male flies were collected after eclosion and groups of 20–40 age-matched flies per genotype were placed together in a vial with fly food. All survival experiments were performed at 29°C. Dead flies were counted daily. Vials were kept on their sides to minimize the possibility of death from flies becoming stuck to the food, and flies were moved to fresh food twice per week. Flies were transferred into new vials without $CO_2$ anaesthesia.

## Negative geotaxis assay/Climbing Assay

Male flies were collected after eclosion and housed for 14 days in age-matched groups of around 20. The assay was performed in the morning, when flies were most active. Flies were transferred without $CO_2$ into a fresh empty vial without any food and closed with the open end of another empty vial. Flies were placed under a direct light source and allowed to adapt to the environment for 20 min. Negative geotaxis reflex was induced by tapping the flies to the bottom of the tube and allowing them to climb up for 8 s. After 8 s the vial was photographed. This test was repeated 3 times per vial with 1 min breaks in between. The height each individual fly had climbed was measured in Image J, and the average between all three runs per vial was calculated.

## Feeding assays

Male flies were aged for 7 days and changed on food supplemented with 0.1% bromophenol blue and 0.5% xylene cyanol. Control flies for each genotype were maintained on non-blue food for background subtraction. Flies were left on the blue or non-blue food for 4 hr at 29°C. After 4 hr flies were anaesthetized with $CO_2$ and decapitated to avoid interference of the eye pigment with the measured absorbance. five flies were homogenized in 100 µl PBS per sample. The fly torsos were homogenized and centrifuged for 20 min at 12.000 rpm. The supernatant was collected and absorbance at 620 nm was analysed with a plate reader.

## Staining of thorax samples

For immunofluorescent staining of thorax muscles, we anaesthetized flies and removed the head, wings and abdomen from the thorax. Thorax samples were pre-fixed for 1 hr in 4% PFA rotating at room temperature. Thoraces were then halved sagitally with a razor blade and fixed for another 30

min rotating at room temperature. Samples were washed with PBS + 0.1% Triton X-100, then blocked for 1 hr in 3% bovine serum albumin (BSA) in PBS + 0.1% Triton X-100.

For Lipid-Tox staining, samples were washed with PBS and stained for 2 hr at room temperature with HCS Lipid Tox Deep Red (Thermo Fisher #H34477; 1:200). For Phalloidin labelling, the samples were washed in PBS after fixation and stained for 2 hr at room temperature with Alexa Fluor 488 Phalloidin (Thermo Fisher #A12379, 1:20). Afterwards the samples were washed once with PBS and mounted in Fluoromount-G. All mounted samples were sealed with clear nail polish and stored at 4° C until imaging.

## Confocal microscopy

Imaging was performed in the Facility for Imaging by Light Microscopy (FILM) at Imperial College London and in the Institute of Neuropathology in Freiburg. A Leica SP5 and SP8 microscope (Leica) were used for imaging, using either the 10x/NA0.4 objective, or the 20x/NA0.5 objective. Images were acquired with a resolution of either $1024 \times 1024$ or $512 \times 512$, at a scan speed of 400 Hz. Averages from 3 to 4 line scans were used, sequential scanning was employed where necessary and tile scanning was used in order to image whole flies. For imaging of whole live flies, the flies were anaesthetized with $CO_2$ and glued to a coverslip. Flies were kept on ice until imaging. For measuring mean fluorescence intensity, a z-stack of the muscle was performed and the stack was projected in an average intensity projection. Next the area of the muscle tissue analyzed was defined and the mean fluorescent intensity within this area was measured. Images were processed and analysed using Image J.

## RNA isolation and reverse transcription

For RNA extraction three whole flies or three thoraces were used per sample. After anaesthetisation, the flies were smashed in 100 µl TRIzol (Invitrogen), followed by a chloroform extraction and isopropanol precipitation. The RNA pellet was cleaned with 70% ethanol and finally solubilized in water. After DNase treatment, cDNA synthesis was carried out using the First Strand cDNA Synthesis Kit (Thermo Scientific) and priming with random hexamers (Thermo Scientific). cDNA samples were further diluted and stored at −20° C until analysis.

## Quantitative Real-time PCR

Quantitative Real-time PCR was performed with Sensimix SYBR Green no-ROX (Bioline) on a Corbett Rotor-Gene 6000 (Corbett). The cycling conditions used throughout the study were as follows: Hold 95°C for 10 min, then 45 cycles of 95°C for 15 s, 59°C for 30 s, 72°C for 30 s, followed by a melting curve. All calculated gene expression values were measured in arbitrary units (au) according to diluted cDNA standards run in each run and for each gene measured. All gene expression values are further normalized to the value of the loading control gene, Rpl1, prior to further analysis.

The following primer sequences have been used in this study:

| Gene name | Forward | Reverse |
| --- | --- | --- |
| Akt1 | 5′-ctttgcgagtattaactggacaga-3′ | 5′-ggatgtcacctgaggcttg-3′ |
| Ilp2 | 5′-atcccgtgattccaccacaag-3′ | 5′-gcggttccgatatcgagtta-3′ |
| Ilp3 | 5′-caacgcaatgaccaagagaa-3′ | 5′-tgagcatctgaaccgaact-3′ |
| Ilp4 | 5′-gagcctgattagactgggactg-3′ | 5′-tggaccggctgcagtaac-3′ |
| Ilp5 | 5′-gccttgatggacatgctga-3′ | 5′-agctatccaaatccgcca-3′ |
| Ilp6 | 5′-cccttggcgatgtatttcc-3′ | 5′-cacaaatcggttacgttctgc-3′ |
| Ilp7 | 5′-cacaccgaggagggtctc-3′ | 5′-caatatagctggcggacca-3′ |
| dome | 5′-cggactttcggtactccatc-3′ | 5′-accttgatgaggccaggat-3′ |
| upd1 | 5′-gcacactgatttcgatacgg-3′ | 5′- ctgccgtggtgctgtttt −3′ |
| upd2 | 5′-cggaacatcacgatgagcgaat-3′ | 5′-tcggcaggaacttgtactcg-3′ |
| upd3 | 5′-actgggagaacacctgcaat-3′ | 5′-gcccgtttggttctgtagat-3′ |

*Continued on next page*

*Continued*

| Gene name | Forward | Reverse |
| --- | --- | --- |
| *Pepck1* | 5'-ggataaggtggacgtgaag-3' | 5'-acctcctgcgaccagaact-3' |
| *Thor* | 5'-caggaaggttgtcatctcgga-3' | 5'-ggagtggtggagtagagggtt-3' |
| *InR* | 5'-gcaccattataaccggaacc-3' | 5'-ttaattcatccatgacgtgagc-3' |
| *Akh* | 5'- agccgtgctcttcatgct-3' | 5'-aaaggttccaggaccagctc-3' |
| *Hsl* | 5'-cttggaaatacttgaggggttg-3' | 5'-agatttgatgcagttctttgagc-3' |
| *bmm* | 5'-gtctcctctgcgatttgccat-3' | 5'-ctgaagggacccagggagta-3' |
| *plin1* | 5'-gcgttctatggtagccttcag-3' | 5'-gcgtccggatagaaagctg-3' |
| *plin2* | 5'-gcagaatggcaagagttctga-3' | 5'-actgtgtgtaggactggatcctc-3' |
| *Rpl1* | 5'-tccaccttgaagaagggcta-3' | 5'-ttgcggatctcctcagactt-3' |

## Smurf assay

Smurf assays with blue-coloured fly food were performed to analyse gut integrity in different genotypes. Normal fly food, as described above, was supplemented with 0.1% Brilliant Blue FCF (Sigma Aldrich). Experimental flies were placed on the blue-coloured fly food at 9AM and kept on the food for 2 hr at 29°C. After 2 hr, the distribution of the dye within the fly was analysed for each individual. Flies without any blue dye were excluded, flies with a blue gut or crop were identified as 'non-smurf' and flies which turned completely blue or showed distribution of blue dye outside the gut were classified as 'smurf'.

## Western blot

Dissected legs or thoraces from three flies were used per sample and smashed in 75 µl 2x Laemmli loading buffer (100 mM Tris [pH 6.8], 20% glycerol, 4% SDS, 0.2 M DTT). Samples were stored at −80°C until analysis. 7.5 µl of this lysate were loaded per lane. Blue pre-stained protein standard (11–190 kDa) (New England Biolabs) was used. Protein was transferred to nitrocellulose membrane (GE Healthcare). Membrane was blocked in 5% milk in TBST (TBS + 0.1% Tween-20). The following primary antibodies were used: anti-phospho(Ser505)-AKT (Cell Signal Technology (CST) 4054, 1:1,000), anti-AKT (CST 4691, 1:1,000), anti-phospho(Thr172)-AMPKα (CST 2535, 1:1,000), anti-phospho(Thr389)-p70 S6 kinase (CST 9206, 1:1,000), anti-GFP (CST 2956, 1:1,000), anti-phospho-p44/42 MAPK (Erk1/2) (CST 4370, 1:1,000) and anti-α-tubulin (clone 12G10, Developmental Studies Hybridoma Bank, used as an unpurified supernatant at 1:3,000; used as a loading control for all blots). Primary antibodies were diluted in TBST containing 5% BSA and incubated over night at 4°C. Secondary antibodies were HRP anti-rabbit IgG (CST 7074, 1:5,000) and HRP anti-mouse IgG (CST 7076, 1:5,000). Proteins were detected with Supersignal West Pico Chemiluminescent Substrate (Thermo Scientific) or Supersignal West Femto Chemiluminescent Substrate (Thermo Scientific) using a LAS-3000 Imager (Fujifilm). Bands were quantified by densitometry using Image J. Quantifications reflect all experiments performed; representative blots from single experiments are shown.

## Thin Layer Chromatography (TLC) for Triglycerides

Groups of 10 flies were used per sample. After $CO_2$ anaesthesia the flies were placed in 100 µl of ice-cold chloroform:methanol (3:1). Samples were centrifuged for 3 min at 13,000 rpm at 4°C, and then flies were smashed with pestles followed by another centrifugation step. A set of standards were prepared using lard (Sainsbury's) in chloroform:methanol (3:1) for quantification. Samples and standards were loaded onto a silica gel glass plate (Millipore), and a solvent mix of hexane:ethyl ether (4:1) was prepared as mobile phase. Once the solvent front reached the top of the plate, the plate was dried and stained with an oxidising staining reagent containing ceric ammonium heptamolybdate (CAM) (Sigma Aldrich). For visualization of the oxidised bands, plates were baked at 80°C for 20 min. Baked plates were imaged with a scanner and triglyceride bands were quantified by densitometry according to the measured standards using Image J.

## Measurement of glucose, Trehalose and Glycogen

5–7-day-old male flies, kept at 29°C, were used for the analysis. Flies were starved for 1 hr on 1% agar supplemented with 2% phosphate buffered saline (PBS) at 29°C before being manually smashed in 75 µl TE + 0.1% Triton X-100 (Sigma Aldrich). three flies per sample were used. All samples were incubated at 75°C for 20 min and stored at −80°C. Samples were thawed prior to measurement and incubated at 65°C for 5 min to inactivate fly enzymes. A total of 10 µl per sample was loaded for different measurements into flat-bottom 96-well tissue culture plates. Each fly sample was measured four times, first diluted in water for calculation of background fly absorbance, second with glucose reagent (Sentinel Diagnostics) for the measurement of free glucose, third with glucose reagent plus trehalase (Sigma Aldrich) for trehalose measurement, and fourth with glucose reagent plus amyloglucosidase (Sigma Aldrich) for glycogen measurement. Plates were then incubated at 37°C for 1 hr before reading with a microplate reader (biochrom) at 492 nm. Quantities of glucose, trehalose and glycogen were calculated according to measured standards.

## Respirometry

Respiration in flies was measured using a stop-flow gas-exchange system (Q-Box RP1LP Low Range Respirometer, Qubit Systems, Ontario, Canada, K7M 3L5). Ten flies from each genotype were put into an airtight glass tube and supplied with our standard fly food via a modified pipette tip. Each tube was provided with $CO_2$-free air while the 'spent' air was concurrently flushed through the system and analysed for its $CO_2$ and $O_2$ content. All vials with flies were normalized to a control vial with food but no flies inside. In this way, evolved $CO_2$ per chamber and consumed $O_2$ per chamber were measured for each tube every ~44 min (the time required to go through each of the vials in sequence).

## Flow cytometry

For flow cytometric analysis of plasmatocytes, 90 flies per sample per genotype were anaesthetized and mechanically dissociated through a 100 µm mesh with 2 mM EDTA in PBS (FACS buffer). The cell suspension is spun down and the resulting cell pellet was resuspended in 5 ml FACS buffer and again rinsed through a 100 µm mesh in a new tube. This washing step was repeated twice. Afterwards the cells were resuspended in 500 µl 2 mM EDTA and Fixable Viability Dye 780 (ebioscience #65-0865-18, 1:1000). Samples were acquired on a FACS Canto II (BD Biosciences) and analyzed with FlowJo analysis software.

## Statistical analysis and handling of data

For real-time quantitative PCR, TLCs, MFI quantification, western blot quantifications and colorimetric measurements for glucose, trehalose and glycogen levels an unpaired t-test or one-way ANOVA was used to calculate statistical significance, as noted in the figure legends. Respirometer data was analysed with a Mann-Whitney test. Lifespan/Survival assays, where analysed with the Log-Rank and Wilcoxon test. Stars indicate statistical significance as followed: *$p<0.05$, **$p<0.01$ and ***$p<0.001$. All statistical tests were performed with Excel or GraphPad Prism software.

All replicates are biological. No outliers were omitted, and all replicates are included in quantitations (including in cases where a single representative experiment is shown). Flies were allocated into experimental groups according to their genotypes. Masking was not used. For survival experiments, typically, the 50% of flies that eclosed first from a given cross were used for an experiment. For smaller-scale experiments, flies were selected randomly from those of a given age and genotype.

## Acknowledgements

We thank the Vienna *Drosophila* RNAi Center, the Bloomington *Drosophila* Stock Center, James Castelli-Gair Hombría, Ernst Hafen, Michael Taylor, Dan Hultmark, Nazif Alic, Bruce Edgar, and the FlyTrap collection at Yale University for flies. We are grateful to Rebecca Berdeaux, Günter Fritz, Marco Prinz, Katie Woodcock, Frederic Geissmann, and members of the South Kensington Fly Room for support, discussion and comments. We thank Maria Oberle for technical assistance. Work in the Dionne lab was supported by funding from BBSRC (BB/P000592/1, BB/L020122/2), MRC (MR/

L018802/2), and the Wellcome Trust (207467/Z/17/Z). KK was supported by a DFG fellowship. JS was supported by BBSRC/GSK CASE studentship BB/L502169/1. FH was supported by the Neuro-mac Graduate School of the SFB/TRR167 and the DFG under Germany's Excellence Strategy (CIBSS-EXC-2189-Project ID 390939984). OG was supported by ERC Starting Grant 337689. The Facility for Imaging by Light Microscopy (FILM) at Imperial College London is part-supported by funding from the Wellcome Trust (104931/Z/14/Z) and BBSRC (BB/L015129/1).

## Additional information

### Funding

| Funder | Grant reference number | Author |
|---|---|---|
| Wellcome | Investigator Award 207467/Z/17/Z | Marc S Dionne |
| Biotechnology and Biological Sciences Research Council | Research Grant BB/P000592/1 | Katrin Kierdorf<br>Pinar Ustaoglu<br>Marc S Dionne |
| Biotechnology and Biological Sciences Research Council | Research Grant BB/L020122/2 | Jessica Sharrock<br>Marc S Dionne<br>Crystal M Vincent |
| Medical Research Council | Research Grant MR/L018802/2 | Katrin Kierdorf<br>Marc S Dionne |
| Deutsche Forschungsge-meinschaft | Research fellowship KI-1876/1 | Katrin Kierdorf |
| Biotechnology and Biological Sciences Research Council | PhD studentship BB/L502169/1 | Jessica Sharrock |
| Deutsche Forschungsge-meinschaft | CIBSS-EXC-2189-Project ID 390939984 | Fabian Hersperger |
| NeuroMac Graduate School of the SFB/TRR167 | | Fabian Hersperger |
| European Commission | ERC starting grant 337689 | Olaf Groß |
| FWF | DASI_FWF01_P29638S | Daria E Siekhaus<br>Attila Gyoergy |
| Medical Research Council | Research Grant MR/R00997X/1 | Crystal M Vincent<br>Marc S Dionne |

The funders had no role in study design, data collection and interpretation, or the decision to submit the work for publication.

### Author contributions

Katrin Kierdorf, Fabian Hersperger, Conceptualization, Formal analysis, Supervision, Funding acquisition, Investigation, Project administration; Jessica Sharrock, Crystal M Vincent, Pinar Ustaoglu, Jiawen Dou, Investigation; Attila Gyoergy, Investigation, Methodology; Olaf Groß, Daria E Siekhaus, Resources, Methodology; Marc S Dionne, Conceptualization, Formal analysis, Supervision, Funding acquisition, Investigation, Methodology, Project administration

### Author ORCIDs

Katrin Kierdorf (iD) https://orcid.org/0000-0002-9272-4780
Jiawen Dou (iD) https://orcid.org/0000-0002-2592-4723
Attila Gyoergy (iD) http://orcid.org/0000-0002-1819-198X
Daria E Siekhaus (iD) http://orcid.org/0000-0001-8323-8353
Marc S Dionne (iD) https://orcid.org/0000-0002-8283-1750

### Decision letter and Author response

Decision letter https://doi.org/10.7554/eLife.51595.sa1
Author response https://doi.org/10.7554/eLife.51595.sa2

## Additional files

### Supplementary files
• Transparent reporting form

### Data availability
Data has been made available on Zenodo, under the DOI https://doi.org/10.5281/zenodo.3608626.

The following dataset was generated:

| Author(s) | Year | Dataset title | Dataset URL | Database and Identifier |
|---|---|---|---|---|
| Kierdorf F, Dionne MS | 2020 | Raw data for Kierdorf et al | https://doi.org/10.5281/zenodo.3608626 | Zenodo, 10.5281/zenodo.3608626 |

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
