## [Decision Letter]

**Acceptance summary:**

The influence of metabolism and lifestyle on muscle function is well known and documented. The role of muscle function in lifespan and metabolism is now, increasingly, being teased out. In this study, the authors investigate the JAK/STAT signalling pathway in *Drosophila* muscle and suggest that it is a critical regulator of lifespan and metabolism. Using genetic interventions, the measurement of lifespan and the analysis of behaviour, the authors show that the genes domeless and hopscotch link to AKT activity to affect metabolic regulation, and lifespan. They also suggest that signals from hemocytes play an important role, though the interpretation of this is not straightforward given the metabolic phenotype is not recapitulated by ablating hemocytes. In sum, the authors make a persuasive case for a link between muscle and lifespan, and the system which they use will increasingly valuable for further studies.

**Decision letter after peer review:**

Thank you for submitting your article "Muscle function and homeostasis require macrophage-derived cytokine inhibition of AKT activity in *Drosophila*" for consideration by *eLife*. Your article has been reviewed by two peer reviewers, and the evaluation has been overseen by K VijayRaghavan as the Senior and Reviewing Editor. The reviewers have opted to remain anonymous.

The reviewers have discussed the reviews with one another and the Reviewing Editor has drafted this decision to help you prepare a revised submission.

Summary:

Kierdorf et al. investigate the JAK/STAT signalling pathway in *Drosophila* muscle as a critical regulator of lifespan and metabolism. The authors show some convincing data linking domeless and hopscotch to AKT activity, metabolic misregulation, and shortened life. Additional data suggest that hemocyte Upd1 is the primary signal. This link is less well supported, but the redundancy between Upd1-3 make these experiments difficult to interpret.

The manuscript needs substantial revision and the authors may want to consider reducing (or even dropping?) the metabolism part.

Essential revisions:

There are a some critical questions that need to be addressed.

1) What is the control line used in these studies? Were all of the fly stocks backcrossed to a parental line? Without backcrossing, the experiments are lacking proper genetic controls. At least for the lifespan studies, the differences observed are easily obtained by heterosis, especially for the lines with multiple transgenic elements. An alternative would have been to use the temperature control, since all studies used a thermogenetic approach, but there appears to be some leakage with the system (Figure 1—figure supplement 1D, although AKT and other metabolic parameters weren't measured). The dependency on only thermogenetic manipulations is a minor weakness, but adding a GeneSwitch or other inducible approach would greatly support the findings.

2) How did the authors determine a single vial of 20 flies would be sufficient to resolve 5% differences in lifespan? And what is a "lifespan effect size"? Do they mean a 5% change in median or mean lifespan? For fly studies, we typically see a requirement of >200 flies to detect 5-10% differences in mean lifespan. Additionally, the various lifespans are described as pooled independent experiments. This doesn't seem right. In Figure 1B, that would mean the experimental line was tested three separate times with less than 20 flies per trial. Individual vials within a single trial should not be considered independent replicates. The lifespan studies are mostly believable, but some of the studies would benefit from independent replication. For example, we don't concur with the conclusion from Figure 4F. The data there is not sufficient to claim a further reduction in longevity with upd2/3 knockouts in a plasmocyte-deleted line.

3) The magnitude of change in AKT levels sometimes seem inconsistent with the effects on lifespan (e.g., comparing Figures 1—figure supplement 1J-K, 1F-H, and Figure 2—figure supplement 1B-C).

4) It is arguable that the metabolic phenotype is due to the haemocyte activity, since it cannot be recapitulated by their ablation.

The authors claim that the phenotype observed upon muscle specific dome inhibition is not due to gut malfunction. They support this claim by assessing gut integrity through the "smurf" assay. However, gut integrity is not equivalent to gut functionality. To properly exclude the gut, the authors should consider crucial gut functions impacting metabolic homeostasis such as feeding, digestion, nutrient absorption or gut contractility. These aspects should be at least discussed.

5) The potential gut contribution to some aspects of the observed phenotype cannot be easily dismissed especially considering its endocrine functions, that can be independent to the gut barrier integrity. In this context, for instance, since dome is expressed by the gut muscular layers, its depletion could potentially trigger upd2 production, which in turn can activate AKH signalling, a known mediator of energy reserve mobilization and positive regulator of basal metabolism.

6) The authors conclude that haemocytes are the mediators of the UPD-dome-STAT-pAKT-foxo circuit in the muscles, since haemocyte ablation reduces STAT-GFP reporter activity. There are two problems here:

Firstly, the extent to which haemocyte ablation is achieved using the experimental conditions is unclear. Can the authors be sure they have removed all hemocytes? Secondly, despite the effects on STAT, hemocyte ablation has no effect on the metabolic phenotype, which implies the presence of an independent, unexplored mediator of dome activation (despite the title of the article which suggests hemocytes as the sole effector of muscle function and homeostasis).

7) The involvement of hemocytes is sustained only by the evidence of stat activation. In this context, a reduced lifespan, alone or in combination with UPDs mutants/IRs is not particularly informative, since it may just reflect the effect of immunosuppression, rather than be related to muscle function and homeostasis. It would be more informative, for example, to assess the effects on climbing activity.

8) While hemocyte ablation, or UPD suppression, reduces STAT-GFP readout, no information is provided on the effect on AKT, which appears to mostly recapitulate the different aspects of muscle dome suppression. If hemocytes are responsible for dome activation, their ablation would phenocopy dome^Δ^ induced upregulation of pAKT in muscles.

If the proposed model is correct, the pAKT upregulation should be rescued by foxo-GFP expression.

9) Another crucial point is the origin of UPD ligands. Only targeted knock-down of all UPDs within the hemocyte population can pinpoint a crucial role for this population on the muscle. The authors should see if this is speedily feasible.

---

## [Author Response]

Essential revisions:There are a some critical questions that need to be addressed.1) What is the control line used in these studies? Were all of the fly stocks backcrossed to a parental line? Without backcrossing, the experiments are lacking proper genetic controls. At least for the lifespan studies, the differences observed are easily obtained by heterosis, especially for the lines with multiple transgenic elements. An alternative would have been to use the temperature control, since all studies used a thermogenetic approach, but there appears to be some leakage with the system (Figure 1—figure supplement 1D, although AKT and other metabolic parameters weren't measured). The dependency on only thermogenetic manipulations is a minor weakness, but adding a GeneSwitch or other inducible approach would greatly support the findings.

We apologise that the genetic backgrounds were not completely clear. We have revised the Materials and methods to clarify this—essentially, all lines were backcrossed onto our isogenic *w1118* control line, except for lines already on another isogenic background that could then be independently compared, and all crosses were performed with driver females so that all the (male) experimental progeny flies would have a uniform X chromosome. This should eliminate the concern regarding heterosis. We agree that a nonthermogenetic system would be desirable as an adjunct to the *Gal80^ts^*approach we have used, but our experience with the GeneSwitch system has been so negative (off-target effects of RU486, weak expression when the system works) that we now avoid its use. However, the manuscript does include non-thermogenetic approaches as well—for example, the Mhc-Gal4 experiments shown in S1.

2) How did the authors determine a single vial of 20 flies would be sufficient to resolve 5% differences in lifespan? And what is a "lifespan effect size"? Do they mean a 5% change in median or mean lifespan? For fly studies, we typically see a requirement of >200 flies to detect 5-10% differences in mean lifespan. Additionally, the various lifespans are described as pooled independent experiments. This doesn't seem right. In Figure 1B, that would mean the experimental line was tested three separate times with less than 20 flies per trial. Individual vials within a single trial should not be considered independent replicates. The lifespan studies are mostly believable, but some of the studies would benefit from independent replication. For example, we don't concur with the conclusion from Figure 4F. The data there is not sufficient to claim a further reduction in longevity with upd2/3 knockouts in a plasmocyte-deleted line.

The comment regarding experimental power is correct—our calculation was wrong because of underestimation of the variance of survival time. (Most of our experience in this regard is with animals dying of bacterial infection, which can show much lower survival variance.) The incorrect statement has been removed. We are fortunate that the survival effect we actually observe in these experiments is generally 25% or more, so that our cohort sizes are large enough for robust conclusions to be drawn. With regard to pooling independent survivals, this statement was in error, due to confusion among the authors regarding the nature of an experimental cohort. With regard to the specific experiment shown in Figure 4F, we have changed our discussion of the result to reflect the fact that we cannot tell whether longevity is further reduced.

3) The magnitude of change in AKT levels sometimes seem inconsistent with the effects on lifespan (e.g., comparing Figures 1—figure supplement 1J-K, 1F-H, and Figure 2—figure supplement 1B-C).

It’s true that the effects we see on lifespan do not map in any obvious, linear way onto the effect on AKT levels (although, in general, manipulations that give smaller effects on AKT tend to give smaller effects on lifespan—the only real exception to this is *Mef2-Gal80^ts^*, which gives a smaller effect on lifespan than would be expected from its effect on AKT abundance). We don’t know why, exactly, beyond the (trivial) observation that dome-AKT signaling is not the only regulator of lifespan. However, our epistasis experiments show clearly that the effect on AKT is an important mediator of the effect of dome inhibition—this is the central point of our work.

4) It is arguable that the metabolic phenotype is due to the haemocyte activity, since it cannot be recapitulated by their ablation.The authors claim that the phenotype observed upon muscle specific dome inhibition is not due to gut malfunction. They support this claim by assessing gut integrity through the "smurf" assay. However, gut integrity is not equivalent to gut functionality. To properly exclude the gut, the authors should consider crucial gut functions impacting metabolic homeostasis such as feeding, digestion, nutrient absorption or gut contractility. These aspects should be at least discussed.

We now discuss these points in the Discussion. We also now provide feeding data for *24BGal80^ts^> dome^Δ^*animals, showing that they consume the same amount of food as the control genotype (Figure 1—figure supplement 1S).

5) The potential gut contribution to some aspects of the observed phenotype cannot be easily dismissed especially considering its endocrine functions, that can be independent to the gut barrier integrity. In this context, for instance, since dome is expressed by the gut muscular layers, its depletion could potentially trigger upd2 production, which in turn can activate AKH signalling, a known mediator of energy reserve mobilization and positive regulator of basal metabolism.

The reviewer is correct about the role of AKH signaling as a known mediator of energy mobilization. We have tested the specific scenario described (upd2 > AKH) by assaying expression of AKH and known and suspected AKH target genes in *24B-Gal80^ts^>dome^Δ^*animals; these data are included as Figure 3—figure supplement 1I-M. In general, though these data are consistent with there being some change in AKH signaling, they do not suggest a strong effect. In a more general sense, we do now discuss other potential contributions of the gut.

6) The authors conclude that haemocytes are the mediators of the UPD-dome-STAT-pAKT-foxo circuit in the muscles, since haemocyte ablation reduces STAT-GFP reporter activity. There are two problems here:Firstly, the extent to which haemocyte ablation is achieved using the experimental conditions is unclear. Can the authors be sure they have removed all hemocytes? Secondly, despite the effects on STAT, hemocyte ablation has no effect on the metabolic phenotype, which implies the presence of an independent, unexplored mediator of dome activation (despite the title of the article which suggests hemocytes as the sole effector of muscle function and homeostasis).

We have now included experimental validation of the plasmatocyte ablation by imaging and flow cytometry (Figure 4—figure supplement 1A, B); we see that almost all (>95%) of plasmatocytes are eliminated. We agree with the reviewer regarding the role of plasmatocytes: we have changed the title to avoid confusion. In the main text we clearly state that plasmatocytes are one relevant source of signal, but they are probably not the only source in wild-type animals (and are certainly not the only source in plasmatocyte-ablated or *upd1*-knockdown animals).

7) The involvement of hemocytes is sustained only by the evidence of stat activation. In this context, a reduced lifespan, alone or in combination with UPDs mutants/IRs is not particularly informative, since it may just reflect the effect of immunosuppression, rather than be related to muscle function and homeostasis. It would be more informative, for example, to assess the effects on climbing activity.

We have assayed climbing activity in plasmatocyte-ablated flies. We found a clear reduction in their climbing activity after hemocyte ablation compared to the control genotype, in line with our findings that hemocyte ablation further results with the appearance of lipid inclusions in muscle of these animals. These data are now included as Figure 4—figure supplement 1C.

8) While hemocyte ablation, or UPD suppression, reduces STAT-GFP readout, no information is provided on the effect on AKT, which appears to mostly recapitulate the different aspects of muscle dome suppression. If hemocytes are responsible for dome activation, their ablation would phenocopy dome^Δ^ induced upregulation of pAKT in muscles.If the proposed model is correct, the pAKT upregulation should be rescued by foxo-GFP expression.

Due to limitations in time we were not able to combine plasmatocyte ablation with the *foxo-GFP* construct. The reviewer is absolutely right that *foxo-GFP* should rescue the high abundance of pAKT; this is shown as Figure 3—figure supplement 1D.

9) Another crucial point is the origin of UPD ligands. Only targeted knock-down of all UPDs within the hemocyte population can pinpoint a crucial role for this population on the muscle. The authors should see if this is speedily feasible.

We agree that this would be desirable, but we have been unable to combine all the required knockdowns in a single fly in the time available to us.